



# Characteristics of earthquake ruptures and dynamic off-fault deformation on propagating faults

Simon Preuss[1], Jean Paul Ampuero[2], Taras Gerya[1], and Ylona van Dinther[3]

[1]Geophysical Fluid Dynamics, Institute of Geophysics, Department of Earth sciences, ETH Zürich, 8092 Zürich, Switzerland
[2]Géoazur Laboratory, Institut de Recherche pour le Développement - Université Côte d'Azur, Campus Azur du CNRS, 06560 Valbonne, France
[3]Tectonics, Department of Earth Sciences, Utrecht University, Princetonlaan 4, 3584 CB, Utrecht, the Netherlands

**Correspondence:** Simon Preuss (sipreuss@ethz.ch)

**Abstract.** Natural fault networks are geometrically complex systems that evolve through time. The evolution of faults and their off-fault damage pattern are influenced by both dynamic earthquake ruptures and aseismic deformation in the interseismic period. To better understand each of their contributions to faulting we simulate both earthquake rupture dynamics and long-term deformation in a visco-elasto-plastic crust subjected to rate-and-state-dependent friction. The continuum mechanics-based nu-
merical model presented here includes three new features. First, a 2.5-D approximation to incorporate effects of a viscoelastic lower crustal substrate below a finite depth. Second, we introduce a dynamically adaptive (slip-velocity-dependent) measure of fault width to ensure grid size convergence of fault angles for evolving faults. Third, fault localization is facilitated by plastic strain weakening of bulk rate-and-state friction parameters as inspired by laboratory experiments. This allows us to for the first time simulate sequences of episodic fault growth due to earthquakes and aseismic creep. Localized fault growth is simulated
for four bulk rheologies ranging from persistent velocity-weakening to velocity-strengthening. Interestingly, in each of these bulk rheologies, faults predominantly localize and grow due to aseismic deformation. Yet, cyclic fault growth at more realistic growth rates is obtained for a bulk rheology that transitions from velocity-strengthening friction to velocity-weakening friction. Fault growth occurs under Riedel and conjugate angles and transitions towards wing cracks. Off-fault deformation, both distributed and localized, is typically formed during dynamic earthquake ruptures. Simulated off-fault deformation structures
range from fan-shaped distributed deformation to localized Riedel splay faults. We observe that the fault-normal width of the outer damage zone saturates with increasing fault length due to the finite depth of the seismogenic zone. We also observe that dynamically and statically evolving stress fields from neighboring fault strands affect primary and secondary fault growth and thus that normal stress variations affect earthquake sequences. Finally, we find that the amount of off-fault deformation distinctly depends on the degree of optimality of a fault with respect to the prevailing but dynamically changing stress field.
Typically, we simulate off-fault deformation on faults parallel to the loading direction. This produces a 6.5-fold higher off-fault energy dissipation than on an optimally oriented fault, which in turn has a 1.5-fold larger stress drop. The misalignment of the fault with respect to the static stress field thus facilitates off-fault deformation. These results imply that fault geometries bend, individual fault strands interact and that optimal orientations and off-fault deformation vary through space and time. With our work we establish the basis for simulations and analyses of complex evolving fault networks subject to both long-term and
short-term dynamics.





## 1 Introduction

Immature strike-slip faults accumulate displacement over time as they undergo a slip localization process. On the long-term, these structures can become deeply penetrating, major faults that represent a highly localized weak zone through the lithosphere (Norris and Toy, 2014). The majority of slip is thereby confined to the cores of the principal faults (Chester et al., 1993). Most

prominent examples, like the San Andreas and the North Anatolian fault systems, span lengths of several hundreds of km (e.g., Sibson, 1983). Strike-slip faults generally grow laterally and the structural fault complexity usually increases towards the younger portions at the fault tip (Perrin et al., 2016a; Cappa et al., 2014). In this area diverse fault patterns and fault networks are found. Analog experiments, structural geology and fracture mechanics define a variety of different secondary fault structure types: branching faults, one-sided horsetail splay faults, synthetic and antithetic Riedel splays, wing cracks and mixed modes

(e.g. Hubert-Ferrari et al., 2003; Kim et al., 2004; Mitchell and Faulkner, 2009; Aydin and Berryman, 2010; Perrin et al., 2016b, Fig. 1 b,c,d). The different types mainly differ in fault angle and fracture mode. For example, R Riedel splays that form in response to Coulomb failure make an angle of $10° - 20°$ with the main fault (Riedel, 1929; Logan. et al., 1979; Logan et al., 1992), while opening-mode wing cracks grow in the direction of the most tensile circumferential stress and hence have a greater angle (Erdogan and Sih, 1963; Ashby and Sammis, 1990). The terms 'shear fracture', 'splay' and 'splay fault' are used

as equivalents in this study. R1 refers to synthetic Riedel shears and R2 refers to antithetic conjugate Riedel shears often also named R'.

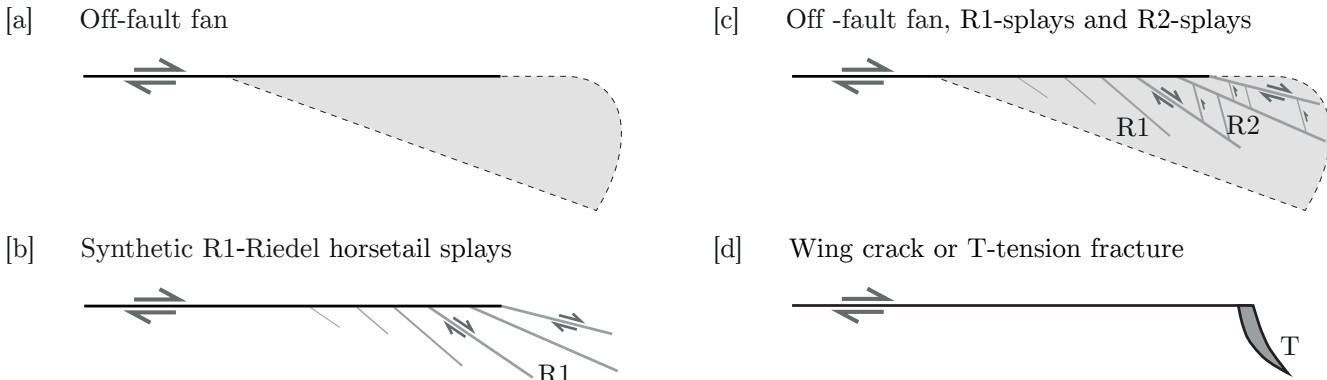

**Figure 1.** Fault structures developing at the tip of a sliding strike-slip fault. Modified from (Kim et al., 2004) and combined with schematic interpretations from (Faulkner et al., 2011) and (Perrin et al., 2016b). R1 = synthetic Riedel shear fracture, R2 = antithetic conjugate Riedel shear fracture and T = tension fracture or wing crack.

The secondary fault structures altogether constitute the wake of a permanent off-fault damage zone (Scholz et al., 1993; Manighetti et al., 2004; Perrin et al., 2016b), alternatively termed off-fault fan (Fig. [1 a,c]). This fan is also present around very well localized principal slip zones (Shipton et al., 2006). Field observations indicate that the damage fan width scales

with accumulated fault displacement (Perrin et al., 2016b, a). At a smaller distance from each fault, an inner damage zone of micro-fractures also develops, whose width does saturate at a few 100 m for larger displacements (Mitchell and Faulkner,





2009; Savage and Brodsky, 2011; Ampuero and Mao, 2017). During an earthquake, energy is dissipated in the damage zone over large distances from the fault (Cappa et al., 2014; Ampuero and Mao, 2017). Earthquakes do not only operate on the main fault structure but can also propagate on secondary faults (Mitchell and Faulkner, 2009). The damage zone also contributes

to long term deformation. For example, secondary faults in California accommodate up to 43% of the total fault slip rate of mapped faults taken from the SCEC catalog (data from Plesch et al., 2017).

Initial attempts to simulate plastic off-fault deformation in elastodynamic earthquake mechanics models were undertaken by Andrews (1976). The plastic fan width was directly related to the rupture propagation distance (Andrews, 2005; Rice et al., 2005). Factors controlling the extent and distribution of off-fault plasticity during dynamic rupture were analyzed by Temple-

ton and Rice (2008). The effect of plasticity on transitions between different rupture styles was studied and the prestress angle was shown to have an influence on rupture speed, plastic zone width and rotation angle of the total seismic moment (Gabriel et al., 2013). A theoretical and numerical study of 3-D rupture with off-fault plasticity revealed how the seismogenic depth of a fault limits the width of the inner damage zone (Ampuero and Mao, 2017). Dynamic earthquake ruptures were shown to propagate along self-chosen paths of a simplified, persistent, branched fault geometry (e.g. Bhat et al., 2004, 2007). Formation

of localization in dynamically generated damage zones was linked to branched faults in a micromechanics based model allowing for the incorporation of crack growth dynamics (Bhat et al., 2012). A recent 2-D dynamic earthquake rupture modeling study nicely shows coseismic off-fault damage during earthquake ruptures at different depths and analyzes its contribution to the overall energy budget (Okubo et al., 2019). Geometrically more complex faults and elastic-plastic off-fault response due to singular events in non-evolving media were studied in a generic case (Fang and Dunham, 2013) or in a realistic fault geometry

model of the Landers earthquake (Wollherr et al., 2019). The main limitation of all these modeling studies is that they are restricted to one single earthquake, a fixed and mostly single main fault that is unable to extend, simplified background stress state represented by a fixed orientation of the principal stress (e.g. Okubo et al., 2019), and an artificial nucleation for numerical convenience (e.g. Gabriel et al., 2013; Okubo et al., 2019). A first attempt to study dynamic branching in self-chosen crack paths was undertaken by Kame and Yamashita (2003). Their study shows crack bifurcation into two branches in homogeneous

elastic media. Their modelling approach does not account for visco-elasto-plastic media and recurrent coseismic events interspersed by aseismic interseismic phases. Recent efforts have advanced the timescales of the simulations to model earthquake cycles on strike-slip faults governed by rate-and-state friction in 2-D antiplane (vertical cross-section) with off-fault plasticity (Erickson et al., 2017). However, 2-D antiplane approaches can not model horizontal propagation of strike-slip faults and their off-fault branches, and models with diffuse plastic deformation can not simulate localized off-fault branches explicitly. Nor

can these dynamic models grow fault branches interseismically.

In this manuscript we develop a computational model that combines the following features: dynamic off-fault yielding in a visco-elasto-plastic material, long-term evolution of a geometrically complicated fault system, consistent simulation of multiple subsequent earthquakes on the same fault system, effect of the finite seismogenic-elastic depth. Our method builds upon and extends the recently developed STM-RSF numerical model for Seismo-Thermo-Mechanical modeling under Rate-

and-State Friction (van Dinther et al., 2013b; Herrendörfer et al., 2018; Preuss et al., 2019) to, for the first time, simulate cyclic seismic and aseismic fault growth. The STM model was developed in van Dinther et al. (2013b, a) to bridge long geodynamic





time scales of fault and lithosphere evolution with short time scales approximating earthquake sequences in a visco-elasto-plastic medium. It was extended to simulate spontaneous off-fault events in van Dinther et al. (2014). Herrendörfer et al. (2018) developed the STM-RSF version to simulate sequences of earthquakes with time steps down to milliseconds using a

new invariant rate-and-state friction (RSF) formulation on predefined, mature faults. Preuss et al. (2019) extended the STM-RSF code to simulate aseismic and seismic fault growth. However, once a fault was formed and ruptured dynamically in a velocity-weakening bulk medium it continued to rupture indefinitely if unhindered by model boundaries.

To accurately simulate cyclic fault growth with off-fault plasticity we extend this STM-RSF framework with three new features. First, we compare four different rate-dependent rheologies in the bulk material. The most realistic one is inspired

by laboratory friction experiments and includes a weakening of RSF parameters with plastic strain (e.g. Beeler et al., 1996; Marone and Kilgore, 1993). These rheologies allow us to simulate distributed and localized coseismic off-fault deformation during dynamic rupture propagation, spanning all the possibilities presented in figure 1. Second, we expand the 2-D framework to 2.5-D using a generalized Elsasser approach (Lehner et al., 1981). Recent work has shown that the thickness of the plate has a direct effect on the width of the inner damage zone leading to a saturation as a function of rupture length (Ampuero and

Mao, 2017). The 2.5-D approximation accounts for stresses generated at depth to counteract sudden displacements originated from a crustal earthquake. It allows us to analyze which factors control the width of the damage zone. Third, we introduce a new rate-dependent fault width formulation to avoid mesh dependency of our simulation results, an issue raised in previous numerical work (e.g. Templeton and Rice, 2008; Bhat et al., 2012). Ultimately, with this new set of models we study the spatio-temporal evolution of an (a)seismically extending pre-existing main fault. Application of driving plate velocities at the

boundaries of the crustal block result in the concentration of stresses on the fault, which leads to consecutive earthquakes. These sudden dynamic events are interspersed by aseismic periods (interseismic phases). The two processes together shape the long-term structure of the fault system by extending the main fault. The mode of extension depends on various properties of the host rock. We compare the orientations of the newly developed faults to the predictions of classical Mohr-Coulomb failure theory and distinguish aseismic from seismic growth contributions. Furthermore, we test the role of the optimality of the angle

of the pre-existing fault on the amount of coseismic off-fault deformation. Various implications of our results are discussed in section 4.

## 2   Methods

We summarize the main ingredients of the STM-RSF modeling approach in section 2.1, but refer the reader to Herrendörfer et al. (2018) for more details. This description focuses on incorporating three new model ingredients: a 2.5-D approximation

(section 2.1), a dynamically adapting fault width during the localization process (section 2.2 ), and a plastic strain dependent bulk rheology behavior as described in the model setup (section 2.3).





## 2.1 Seismo-mechanical modeling with rate- and state-dependent friction and a 2.5-D approximation

We expand the 2-D STM-RSF framework to 2.5-D using a generalized Elsasser approach introduced by Rice (1980); Lehner et al. (1981). The 2.5-D model captures the concept that rapid deformation in the elastic-brittle upper crust exerts a shear stress

load onto the deeper visco-elastic crustal substrate, which then relaxes slowly and transfers stresses back to the upper crust (Lehner et al., 1981). Assuming, for simplicity, a Maxwell coupling between the upper and lower crust, the shear tractions at their interface are approximated as thickness-averaged stresses (ch. 10 in Rice, 1980). Weng and Ampuero (2019) showed that the energy release rate of dynamic ruptures in a 2.5-D model approximates very well that of long ruptures with finite seismogenic depth in a 3-D model. The 2.5-D simulations thus approximately account for the 3-D effect of a finite rupture

depth at the same computational cost as a 2-D simulation.

In 2.5-D we solve for the conservation of mass:

$$\rho \frac{\partial v_i}{\partial x_i} = -\frac{D\rho}{Dt},\tag{1}$$

and the conservation of momentum:

$$\frac{\partial \tau_{ij}}{\partial x_j} - \frac{\partial P}{\partial x_i} = \rho \frac{Dv_i}{Dt} - \rho g_i,\tag{2}$$

where $i = 1, 2$ and $j = 1, 2, 3$ are coordinate indices, $x_i$ and $x_j$ represent spatial coordinates, $\rho$ denotes density, $\frac{D}{Dt}$ is the material time derivative, $v_i$ is velocity and $g_i$ is gravity. $P$ is the dynamic pressure, defined as positive under compression, and

computed from the mean stress as:

$$P = -\frac{\sigma_{kk}}{3}, \qquad \text{with } k = 1, 2, 3,\tag{3}$$

The deviatoric stress tensor $\tau_{ij}$, with $i = j = 1, 2$, in the crust is given as:

$$\tau_{ij} = \sigma_{ij} + \delta_{ij} P,\tag{4}$$

with $\sigma_{ij}$ being the Cauchy stress tensor and $\delta_{ij}$ the Kronecker delta. They are linked to deviatoric strain rates $\dot{\varepsilon}'_{ij}$ by the following visco-elasto-plastic constitutive relationship (Gerya and Yuen, 2007):

$$\dot{\varepsilon}'_{ij} = \frac{1}{2G} \frac{\overset{\triangledown}{D}\tau_{ij}}{Dt} + \frac{1}{2\eta}\tau_{ij} + \dot{\varepsilon}'_{II(\text{plastic})} \frac{\tau_{ij}}{\tau_{II}},\tag{5}$$

where $G$ is the shear modulus, $\frac{\overset{\triangledown}{D}}{Dt}$ denotes the co-rotational time derivative, $\eta$ is the effective ductile viscosity in the crust,

$\dot{\varepsilon}'_{II(\text{plastic})}$ is the second invariant of the deviatoric plastic strain rate and $\tau_{II} = \sqrt{\tau_{11}^2 + \tau_{12}^2 + \tau_{13}^2 + \tau_{23}^2}$ is the second invariant of the deviatoric stress tensor in 2.5-D. The stresses at depth are averaged over the thickness of the crust $T$ as (Lehner et al., 1981):

$$\tau_{i3} = \frac{G_\text{S}\, u_i}{bT} + \frac{\eta_\text{S}\, \dot{u}_i}{T_\text{S}},\tag{6}$$





where $u_i$ are the displacements averaged over the thickness of the crust, $T$, and $T_\text{S}$ is the thickness of the lower crustal substrate with shear modulus $G_\text{S}$ and viscosity $\eta_\text{S}$, respectively. The geometric factor $b = 1/\pi^2$ is designed to match the energy release

rate between 2.5-D and 3-D dynamic rupture simulations (Weng and Ampuero, 2019). The medium is compressible, so density and pressure are related by

$$\frac{D\rho}{Dt} = \frac{\rho}{K}\frac{DP}{Dt}, \tag{7}$$

where $K$ is the bulk modulus. The onset of plastic deformation is defined by the yield criterion $F = 0$ with a Drucker-Prager plastic yield function (Drucker and Prager, 1952):

$$F = \tau_{II} - \sigma_{yield} \tag{8}$$

with local pressure-dependent yield strength

$$\sigma_{yield} = \mu_l(1-\lambda)P + C, \tag{9}$$

where $\mu_l$ is the local friction coefficient (it can vary both in space and time), $C$ denotes cohesion (residual shear strength at zero effective pressure $P$), and $\lambda$ is the pore fluid pressure factor (ratio between fluid pressure and solid pressure). The local friction coefficient evolves according to the invariant reformulation of rate- and state-dependent friction for a continuum, introduced by Herrendörfer et al. (2018). This formalism was applied to freely and spontaneously growing seismic and aseismic faults by Preuss et al. (2019), by interpreting how plastic deformation starts to localize and forms a shear band that approximates a fault

zone of finite width that can host earthquakes. Localized bulk deformation and fault slip are related by defining the plastic slip rate $V_\text{p}$ as

$$V_\text{p} = 2\dot{\varepsilon}'_{II(p)}W, \tag{10}$$

where $W$ denotes the width of the fault zone in the continuous host rock. We write equation 9 as:

$$\sigma_{yield} = \mu_l(1-\lambda)P = a\,\text{arcsinh}\left[\frac{V_\text{p}}{2V_0}\exp\left(\frac{\mu_0 + \frac{C}{P} + b\ln\frac{\theta V_0}{L}}{a}\right)\right]P(1-\lambda), \tag{11}$$

where $a$ and $b$ are laboratory-based, empirical RSF parameters that quantify a direct effect and an evolution effect of friction, respectively, $L$ is the RSF characteristic slip distance, $\mu_0$ is a reference friction coefficient at a reference slip velocity $V_0$

(Lapusta and Barbot, 2012), and $C$ is the cohesion as part of the state variable $\theta$ (Marone et al., 1992) that evolves according to the aging law:

$$\frac{d\theta}{dt} = 1 - \frac{V_\text{p}\theta}{L}. \tag{12}$$

To solve the governing equations we use an implicit, conservative finite-differences scheme on a fully staggered grid combined with the marker-in-cell technique (Gerya and Yuen, 2003, 2007). All details of the numerical technique that comprise the STM-RSF code can be found in Herrendörfer et al. (2018).



## 2.2 Slip-velocity-dependent fault-width formulation


In contrast to previous studies, in which fault width $W$ in the slip-rate formulation of friction was constant (e.g., van Dinther et al., 2013b, a; Herrendörfer et al., 2018; Preuss et al., 2019), we introduce here a dynamically adapting $W$. Deformation localizes to within 1-2 grid cells in classical applications of plasticity (e.g., Lavier et al., 2000; Buiter et al., 2006; van Dinther et al., 2013a). In models involving a pre-existing fault, Herrendörfer et al. (2018) found that setting $W = \Delta x$ leads to conver-

gence with respect to grid size. However, this choice is not optimal in models involving the spontaneous formation of a fault, because it leads to grid-size-dependent fault orientations (on the order of a few degrees) and earthquake onset times (Preuss et al., 2019). This results from the elastic loading phase, in which, for larger grid sizes, the same slip velocity is scaled to a smaller effective plastic strain rate. This yields a higher visco-plastic viscosity and thus, higher stresses, resulting in a larger slip velocity. This higher slip velocity, in turn, leads to a faster evolution of state and thus, to a faster localization of deformation.

The higher stresses additionally induce a higher local friction coefficient in the undeformed matrix and hence, the fault angle becomes larger (Preuss et al., 2019). To prevent this dependence on grid size, we introduce a length scale in the relationship between slip rate and plastic strain rate that incorporates the distributed deformation during the fault localization phase, which may span more than one grid cell.

We propose a new invariant continuum-based RSF formulation, in which the fault width $W$ adapts dynamically as a function

of the slip velocity $V_p$ during the strain localization phase. We define a rate-dependent width

$$W_{V_\mathrm{p}} = W_\mathrm{max} \, \log \left( 1 + K \frac{V_0}{V_p} \right) \tag{13}$$

and complement it with lower and upper bounds as

$$W = \begin{cases} W_\mathrm{max}, & \text{if } W_{V_\mathrm{p}} \geq W_\mathrm{max} \\ W_{V_\mathrm{p}}, & \text{if } W_\mathrm{max} \geq W_{V_\mathrm{p}} \geq \Delta x \\ \Delta x, & \text{if } W_{V_\mathrm{p}} \leq \Delta x \end{cases} \tag{14}$$

The dimensionless scaling parameter $K = 1$ determines the onset of localization. We set the upper fault width limit $W_\mathrm{max} = 20$ km based on averaged values from GPS and InSAR data across several natural faults. The above relation can be interpreted as a heuristic fix to the problem of grid-size-dependent localization in continuum models with RSF. We base this fix on the

fact that slow crustal movements in the interseismic period lead to distributed deformation (Collettini et al., 2014), e.g., within the Pacific-North America plate boundary, which is very wide in the north, across the San Andreas fault and the Eastern California shear zone segments (Wdowinski et al., 2007). On the other hand, faults show localization of deformation during coseismic periods, sometimes identifiable at large depths by the presence of pseudotachylyte (Sibson, 1977, 1983; Swanson, 1992; Norris and Toy, 2014; Ikari, 2015). Examples of highly localized slip planes are numerous, in different rock types

and on different scales (e.g., Chester et al., 1993; Chester and Chester, 1998; Sibson, 2003; Smith et al., 2011; Barth et al., 2013; Collettini et al., 2014; Rice, 2017). Also laboratory experiments reveal that strain is initially distributed across the full thickness of a sheared gouge layer, but after some displacement it localizes to a high-strain principal slip surface (Smith et al.,





2015; Ritter et al., 2018a, b). These principal slip surfaces, which seem to be a prerequisite for future earthquake slip, form at subseismic slip velocities (Ikari, 2015). This behavior is affirmed by numerous experimental and micro-structural studies

that report a fine-grained shear localization zone cut by a discrete principal slip zone during sliding at seismic velocities (e.g., Toro et al., 2004; Brantut et al., 2008; Han et al., 2010; Scuderi et al., 2013; Paola et al., 2015; Pozzi et al., 2018). Alloy in extension and compression shows an analogous sequence of formation of coarse deformation bands followed by shear strain localization leading to crack formation (Price and Kelly, 1964). A common mode of failure in granular materials involves the same pattern of localization of homogeneous deformation into a narrow zone of intense shear (Jenkins, 1990). In addition,

our heuristic fix follows the general tendency for logarithmic rate-dependency of materials, which appears in, e.g., friction and porosity/dilatancy (Dieterich, 1981; Logan and Rauenzahn, 1987; Sleep, 1995; Segall and Rice, 1995; Beeler et al., 1996; Chen et al., 2017).

The result of using the new slip-velocity-dependent fault-width formulation is that both the fault angle and the onset times of earthquakes converge with grid size (appendix A).

## 2.3 Model setup

Our 2.5-D model setup represents a generic case to study the evolution of a fault zone within a plane-strain crust of 20 km thickness coupled to an underlying lower crustal layer of 20 km thickness. The model horizontal size is 250 km x 150 km. The third dimension is reduced by depth-averaging, leading to a 2.5-D model (section 2.1). The initial fault is prescribed as a 120 km long line of lower state $\theta$ and zero cohesion $C$ compared to the host rock. It represents the tip of a mature fault in nature

and will be the locus of stress concentration leading to the spontaneous nucleation and propagation of a rupture as a mode II crack. The experimental geometry, together with the Dirichlet $v_x$-velocity boundary conditions applied in opposite directions at the back and front boundaries, represent a dextral strike slip zone (Fig. [2]).

The maximum compressive stress $\sigma_1$ is initially oriented at an angle of $45°$ to the imposed shear direction, indicated by the light green arrows in Fig. [2] (e.g., Meyer et al., 2017). Values of the reference model parameters (Table 1) are set largely

in accordance with Lapusta et al. (2000), with differences in the choice of $V_0$ and the initial mean stress $P_B$. Following Herrendörfer et al. (2018), we set $V_0$ equal to the loading rate. The background effective pressure $P_B = 20$ MPa can be related to the lithostatic pressure $P_{B_{\text{lith}}}$ and the pore fluid pressure $P_f$ by

$$P_B = P_{B_{\text{lith}}} - P_f = P_{B_{\text{lith}}}(1 - \lambda). \tag{15}$$

Considering a typical value of pore fluid pressure ratio $\lambda \sim 0.67$, the lithostatic pressure is $P_{B_{\text{lith}}} = 60.6$ MPa, which is equivalent to a depth of 2.3 km, representing the upper crust. In section 3.2.1 we simulate at higher and lower pressures that represent

higher and lower depths, respectively.

In this study we present four different reference models with varying RSF behavior in the bulk: model *RS* has rate-strengthening behavior, model *RN* has rate-neutral behavior, model *RW* has rate-weakening behavior, and model *RT* has a transition from rate-strengthening to rate-weakening behavior at increasing plastic strain. We choose to study these different bulk behaviors because in the literature of materials and geology both strain-rate-strengthening and strain-rate-weakening are



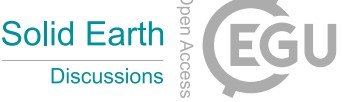

**Table 1.** Rate- and state-dependent friction (RSF) and material parameters of the four 2.5-D reference models

| Parameters | Symbol | Value |
|---|---|---|
| **Upper Crust** | | |
|     Thickness | $T$ | 20 km |
|     Shear modulus | $G$ | 50 GPa |
|     Bulk modulus | $K$ | 50 GPa |
|     Density | $\rho$ | 2700 kg/m$^3$ |
|     Shear wave speed | $c_s$ | 4.3 km/s |
|     Effective viscosity | $\eta$ | $5 \cdot 10^{26}$ Pa s |
|     Initial mean stress (pressure) | $P_B$ | 20 MPa |
|     Gravity | $g_i$ | 0 m/s$^2$ |
|     Reference effective-static friction coefficient | $\mu_0$ | 0.6 |
|     Cohesion | | |
|         Host rock | $C_{\mathrm{hr}}$ | $6 \cdot 10^6$ MPa |
|         Fault | $C_{\mathrm{f}}$ | 0 MPa |
|     Initial state | | |
|         Host rock | $\theta_{\mathrm{hr}}$ | $\frac{L}{V_0} \exp(30)$ s $\approx 8.5 \times 10^8$ years |
|         Fault | $\theta_{\mathrm{f}}$ | $\frac{L}{V_0} \exp(-1)$ s $\approx 0.03$ years |
|     Reference slip velocity | $V_0$ | $4 \cdot 10^{-9}$ m/s = 12.6 cm/yr |
| **Reference model rate-strengthening *RS*** | | |
|     Characteristic slip distance | | |
|         Host rock | $L_{\mathrm{hr}}$ | 1.0 m |
|         Fault | $L_{\mathrm{f}}$ | 0.01 m |
|     RSF direct effect | $a$ | 0.011 |
|     RSF evolution effect | | |
|         Host rock | $b_{\mathrm{hr}}$ | 0.007 |
|         Fault | $b_{\mathrm{f}}$ | 0.017 |
| **Reference model rate-neutral *RN*** | | |
|     Characteristic slip distance | | |
|         Host rock | $L_{\mathrm{hr}}$ | 0.01 m |
|         Fault | $L_{\mathrm{f}}$ | 0.01 m |
|     RSF direct effect | $a$ | 0.011 |
|     RSF evolution effect | | |
|         Host rock | $b_{\mathrm{hr}}$ | 0.011 |
|         Fault | $b_{\mathrm{f}}$ | 0.011 |
| **Reference model rate-weakening *RW*** | | |
|     Characteristic slip distance | | |
|         Host rock | $L_{\mathrm{hr}}$ | 0.01 m |
|         Fault | $L_{\mathrm{f}}$ | 0.01 m |
|     RSF direct effect | $a$ | 0.011 |
|     RSF evolution effect | | |
|         Host rock | $b_{\mathrm{hr}}$ | 0.017 |
|         Fault | $b_{\mathrm{f}}$ | 0.017 |
| **Reference model rate-transitioning *RT*** | | |
|     Characteristic slip distance | | |
|         Host rock | $L_{\mathrm{hr}}$ | $1.0 \rightarrow 0.01$ m (if $\varepsilon_p\ 0 \rightarrow$ 5e-4) |
|         Fault | $L_{\mathrm{f}}$ | 0.01 m |
|     RSF direct effect | $a$ | 0.011 |
|     RSF evolution effect | | |
|         Host rock | $b_{\mathrm{hr}}$ | $0.007 \rightarrow 0.017$ (if $\varepsilon_p\ 0 \rightarrow$ 5e-4) |
|         Fault | $b_{\mathrm{f}}$ | 0.017 |
| **Lower crustal substrate** | | |
|     Thickness | $T_{\mathrm{S}}$ | 20 km |
|     Shear modulus | $G_{\mathrm{S}}$ | 50 GPa |
|     Viscosity | $\eta_{\mathrm{S}}$ | $1 \cdot 10^{17}$ Pa s |



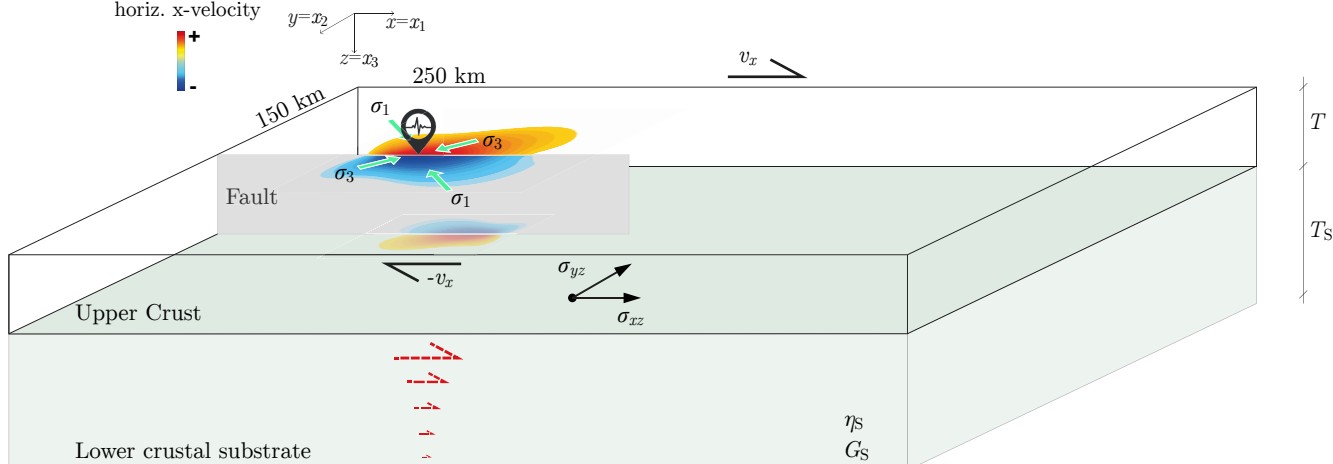

**Figure 2.** 2.5-D model setup of the dextral inplane strike slip simulation: 2-D box of size 250 km x 150 km with 501 x 301 nodes in x- and y-direction, respectively (grid resolution of 500 m). Each cell contains 16 markers, resulting in 2,400,000 global mobile markers. Bold black shear-arrows indicate direction of Dirichlet $v_x$-velocity boundary conditions applied in opposite direction: $v_x = \pm 2 \cdot 10^{-9}$ m/s $= \pm 6.4$ cm/yr. At left and right boundaries, Neumann boundary conditions for $v_x$ are prescribed. Velocity $v_y$ is set to zero at all boundaries. The 120 km long fault (in gray) has a lower state $\theta$ compared to the host rock. The third ($z$) dimension is approximated by a depth-averaging of upper crustal stresses due to a relaxed lower crustal substrate (section 2.1). A symbolic earthquake rupture occurring on the fault is shown by horizontal x-velocity contours in red-blue at the location of a seismic wave sign. Light green arrows with white contour mark the direction of the principal compressional stresses, the maximum one, $\sigma_1$, is initially oriented at $45°$. The counteracting effect of the rupture on stresses in the relaxed substrate is shown by the red dashed arrows. The directionality of shear tractions $\sigma_{xz}$ and $\sigma_{yz}$ on the lower surface of the upper crust is represented by black arrows with triangle heads. Quantities $\eta_S$, $G_S$ and dimensions $T$ and $T_S$ are listed in table 1.

reported to be possible and sufficient conditions for localization of deformation (Hobbs et al., 1990). By studying these different bulk behaviors we are able to emphasize their different characteristics. In the first three cases, $(a - b)$ and $L$ are kept constant in the entire model domain, respectively. In model RT, $(a - b)$ and $L$ decrease with plastic strain (Table 1), motivated by laboratory observations (Beeler et al., 1996; Scuderi et al., 2017; Marone and Kilgore, 1993).

## 3    Results and analysis

In the first part of this section we present and compare the results of the four models to understand the effects of a rate-sensitive bulk rheology on off-fault deformation and fault growth. We then focus on two reference models to further investigate factors influencing the off-fault fan width and the role of visco-elastic lower crust relaxation on short-term and long-term off-fault deformation (section 3.2). In section 3.3 we study the role of the angle of the predefined fault on off-fault deformation. Finally, we analyze the formation of localized secondary branches (section 3.4).





## 3.1 Role of rate-dependent bulk rheology on off-fault deformation and fault growth

The initial elastic loading phase takes 325.5 years in all four reference models. In this initial stage stresses start to localize on the predefined fault and its slip velocity increases exponentially (Fig. [3b]). About 5.5 years after the slip velocity reaches the plate velocity, an earthquake nucleates on the fault at x=0 km and propagates eastwards along the fault. During the earthquake, the on-fault slip rate $V_p$ rises to seismic slip velocities, as defined by the typical dynamic scale $V_{\mathrm{seis}}$ introduced by Rubin and Ampuero (2005) (Fig. [3b]). The generation of coseismic off-fault plastic deformation is a general feature of our models (Fig. [3a]). It occurs on the extensional side of the fault, which is favored over the compressional side when the maximum principal compressional stress direction is $\geq 45°$ (Kame et al., 2003). The off-fault plastic zone grows continuously in a fan-like manner but eventually its thickness (fault-normal size) saturates at a value of $\sim$8 km and at a rupture distance of $\sim$80 km. Concurrently, the slip velocity reaches a maximum at $V_p = 1.1$ m/s. The maximum width of the off-fault fan $H_{\mathrm{max}}$ and the onset of saturation are affected by various parameters investigated in section 3.2.1.

Besides these general similarities among the four reference models we note two main differences. First, the plastic zones off the main fault have distinct characteristics. A fan of diffuse deformation occurs in models RS and RT, while localized deformation on secondary faults occurs in models RN and RW. Thus, the type of plastic off-fault yielding depends on the properties of the bulk rheology. Only a rate-neutral or rate-weakening material in which $L$ is comparatively low (0.01 m) allows for fault localization in the off-fault material. Each of the secondary fault branches is formed by individual secondary dynamic ruptures during the first main fault rupture. This localization behavior will be addressed further in section 3.4.

The second main difference is that only model RT hosts a sequence of several earthquakes. In the other three models the entire new fault geometry forms during a single earthquake, an aftershock and the subsequent post-seismic and inter-seismic phases. Before more seismic events can nucleate, the main fault has already extended by aseismic growth toward a model boundary and the simulation is stopped to prevent boundary effects. Thus, models RS, RN, and RW have in common that fault growth during the earthquakes contributes only a small portion to the total formed fault length $F_L$ (Fig. [3b]). Furthermore, their fault growth rates of up to 77 km/yr are much faster than in model RT. We follow that a bulk rheology with constant rate-sensitivity favors a faster fault growth. In contrast, the heterogeneities introduced by a weakening of the RSF parameters $L$ and $b$ slow down the faulting process due to the absorption of energy by the weakening mechanism. As a consequence, the faults in model RT can extend in alternating seismic and aseismic growth periods. Only if the region ahead of the fault tip has experienced distinct plastic strain and $L$ and $b$ are altered to create a rate-weakening fault earthquakes can propagate on there. Otherwise, dynamic rupturing is hindered in the intact bulk, where $L$ is still high and $b$ is still low and rate-strenghtening, respectively. This results in intermittent seismic and aseismic growth sequences. We think this behaviour reflects the natural growth of crustal faults better. This difference has major implications on the dynamics and geometry of fault evolution, as discussed in the next section. Nevertheless, all reference models have in common that fault growth during the earthquakes contributes only a small portion to the total formed fault length $F_L$ (Fig. [3b]).

In the following analysis we focus on models RW and RT because the off-fault characteristics of models RW and RN are similar, as well as those of models RT and RS. Furthermore, RW and RT represent the two cases of dominant higher-angle





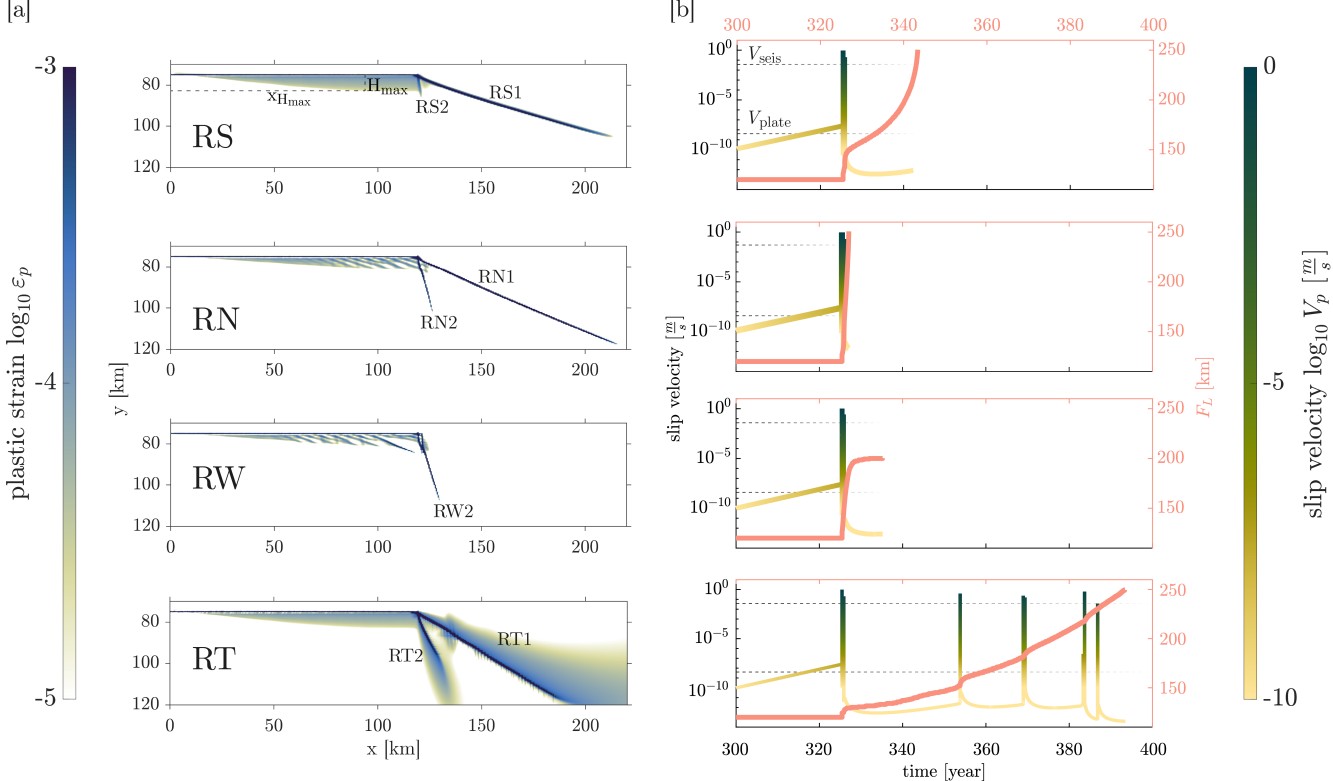

**Figure 3.** Summary of four different right-lateral reference models. [a]: Simulation results in form of plastic strain pattern: RS= rate-strengthening, RN= rate-neutral, RW = rate-weakening, RT = transition from RS to RW. 'R*1' indicates a Riedel fault (lower strike angle), 'R*2' refers to the conjugate Riedel fault (R') with a higher strike angle compared to the R1-Riedel shear, respectively. [b]: Temporal evolution of the plastic slip velocity $V_p$ inside the fault at $(x, y) = (100\,\text{km}, 75\,\text{km})$ until end of simulation. The seismic threshold velocity is indicated as $V_{\text{seis}}$. Red line shows the fault length $F_L$ over time.

and lower-angle continuing faults, respectively. In that way, RW and RT are the most diverse end member cases of the four
reference models. At the same time model RT is the most realistic model with successive earthquakes, distributed deformation
and localized fault growth. Model RW on the other hand allows for strong off-fault localization and tends towards irregular
fault patterns with unevenly spaced secondary fault branches.

### 3.1.1   Temporal and geometrical evolution of fault growth

As the main fault propagates beyond the tip of the predefined fault, all four reference models form two new faults with two
orientations. A "higher-angle fault" forms with a high angle compared to the strike of the predefined fault. It is an antithetic
conjugate Riedel shear fault and is termed 'R*2', where '*' stands for the different reference models. A "lower-angle fault"
forms with a lower strike angle and is a synthetic Riedel shear fracture termed 'R*1' hereafter. Detailed analysis of model





RT reveals that these two faults are formed because the main fault rupture induces two stress lobes on the extensional side of the fault. These zones were studied by Poliakov et al. (2002) who termed them 'secondary faulting area'. These areas also

evoke lobe-like anomalies of the dynamic friction value. Later on, the frontal lobe is responsible for the formation of RT1 and the back lobe forms the conjugate fault RT2 (Fig. [4a,b]). This behavior of model RT is representative for all four reference models. We confirm that both faults form according to the Mohr-Coulomb failure criterion at either the Coulomb-fault angle (R1 fault) or the conjugate angle (R2 fault). These angles $\alpha$ are

$$\alpha = \pm \left( \frac{\pi}{4} \mp \frac{\varphi}{2} \right), \tag{16}$$

where $\varphi$ denotes the internal angle of friction, related to $\mu_l$, the local friction coefficient, by $\mu_l = \tan\varphi$. The angle $\alpha$ spans

between the maximum principal compressional stress direction $\sigma_1$ and $\beta$, the angle of the newly forming fault, such that $\alpha = \sigma_1 - \beta$ (Fig. 4c). The reference angle is the horizontal shear direction. The compressive state of stress in our simulation limits $\alpha$: $\alpha_1 = 45° - (\varphi/4)$ and $\alpha_2 = -45° + (\varphi/4)$ (conjugate). We take the average of $\sigma_1$ and $\mu_l$ to compute the resulting $\alpha$ at 3 different instants: 1) five seconds before fault extension, 2) one minute after the initiation of the first fault extension, and 3) 25 years after the initial fault extension (Fig. [4d], symbols ■, ●, ♦, respectively). These data reveal that the local

dynamic conditions in the proximity of faults determine the angles of the faults RT1 and RT2, which was previously reported for an isolated growing fault (Preuss et al., 2019). Additionally, they approach the theoretical angle $\alpha$ and converge on it with advancing time (Fig. [4d]). This behavior indicates that the absolute fault angles $\beta$ of RT1 and RT2 are predictable, even before the faults have formed. As soon as the two lobes are formed by elevated stresses near the rupture front, i.e. approximately 10 seconds after the earthquake nucleation, they have the potential to determine the angle of the R-fault and the R2-conjugate

(Fig. [4e]). Both RT1 and RT2 exhibit a lower final strike angle than their predictions during the dynamic phase. This is justified by the fact that seismically growing faults have a higher strike angle than aseismically forming faults (Preuss et al., 2019). Thus, the seismically initiated fault decreases in strike angle as soon as the slip velocity transitions to the non-seismic range. This happens as the rupture hits the undeformed host rock. Indeed, at the end of the first earthquake the fault extends in a pure seismic mode and the angle of the new fault is greater than in the following interseismic phase after the earthquake.

Hence, both RT1 and RT2 flatten in the 0.3 years right after the first earthquake and before the aftershock. This behavior is visible in the online videos (link to video RT, 1.1 Gb, RS, 207 Mb, RN, 204 Mb, RW, 170 Mb) just after the first earthquake, in Fig. 4b and in all four reference model snapshots (Fig. [3a]).

The different reference models favor either of the two fault angle types or develop them jointly. The smaller the initial $L$ and $a - b$ in the bulk material, i.e. with more weakening, the higher the tendency to favor the high-angle conjugate fault (Fig. 3).

This behavior is evident if one compares models RS and RW. An intermediate situation occurs in models RN and RT when both fault types are well developed and evolve in an approximately equal manner. A second result of lower initial $L$ and $a - b$ is the greater generated fault length in response to the primary seismic event: $F_L = 12.3$ km (RW2), 11.3 km (RN2), 8.3 km (RS2) and 7.4 km (RT2). Hence, an earthquake can extend a fault farther in a rate-weakening material than in a rate-strengthening material, and the rate-transition case produces the shortest seismic growth. The aseismic fault growth rate in a rate-neutral



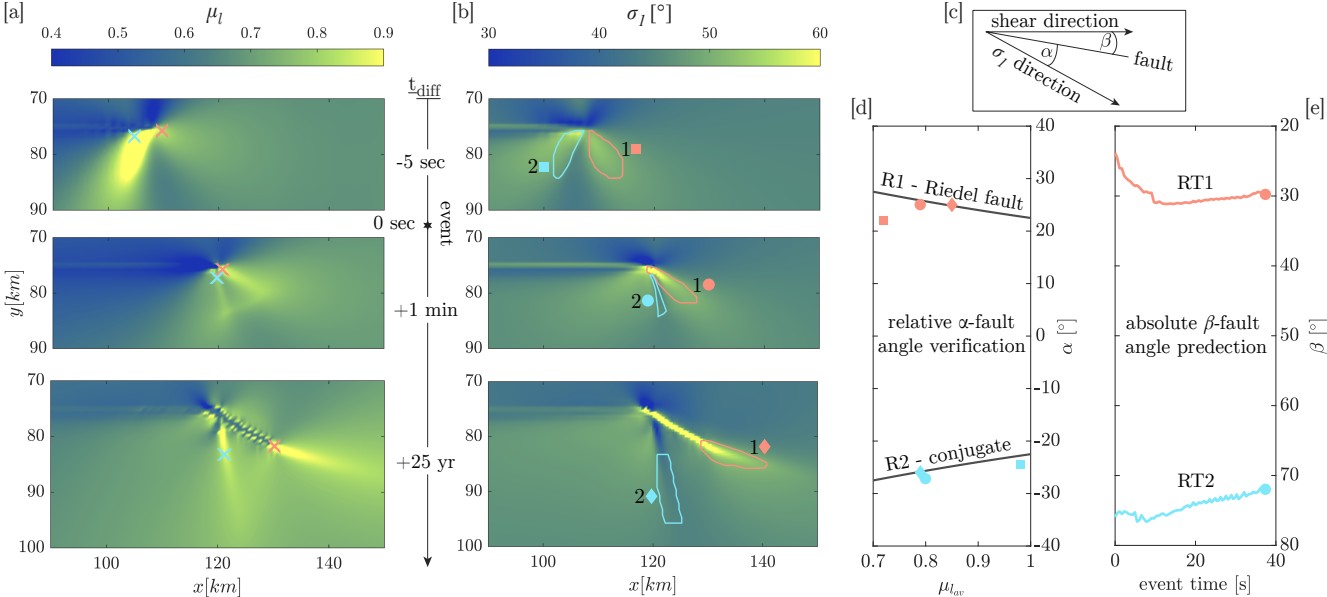

**Figure 4.** Evolution of friction coefficient and stress orientation over time and prediction of the absolute fault angle. 3 different times are: 1) Five seconds before fault extension (symbol ■), 2) One minute after the initiation of the first fault extension (symbol ●), and 3) 25 years after the initial fault extension (symbol ♦). Red color represents the Coulomb angle of R1 and turquoise refers to the conjugate fault R2. [a]: Snapshot zooms of the local friction distribution $\mu_l$ at different times. Crosses indicate the location of the two maxima of $\mu_l$ at the rupture tip or at the tip of the newly formed fault, respectively. [b]: Snapshot zooms of the distribution of the maximum compressive stress orientation $\sigma_1$ at different times. Red and turquoise polygons indicate the locations around the frictional maxima inside the two stress lobes where $\mu_l$ and $\sigma_1$ are sampled and averaged according to two standard deviations of the mean (95th percentile). [c]: Schematic geometrical relation in physical space between horizontal shear direction, $\sigma_1$-direction and forming fault. $\beta$ represents the absolute fault angle, $\alpha$ is the angle between fault and $\sigma_1$-direction, such that $\alpha_{1,2} = \sigma_1 - \beta_{1,2}$. [d]: Relation between averaged friction $\mu_{l_{av}}$ and fault angle $\alpha_{1,2}$, which is obtained from the average $\sigma_1$-directions for the three different instants shown in [a] and [b] for R1 and R2, respectively. The values from the frontal lobe result in the data converging on the R1-Riedel angle $\alpha_1$, the data from the back lobe converge on the R2-conjugate $\alpha_2$. Black lines indicate the theoretical angle given by equation 16. [e]: Prediction of the absolute fault angles $\beta_{1,2}$ for both RT1 and RT2 from sampling averages of $\mu_l$ and $\sigma_1$ during the entire first dynamic earthquake.

material is higher than in a rate-strengthening material (Fig. [3b]). Most likely it is even higher in a rate-weakening material but a comparison is impeded because model RW favors a different fault growth path.

In the following we describe the evolution of the R1-fault and the conjugate R2 in detail for the four individual model cases (link to video RT, 1.1 Gb, RS, 207 Mb, RN, 204 Mb, RW, 170 Mb). In the RS model with pure rate-strengthening behavior RS1 and RS2 are created simultaneously as the seismic rupture penetrates into the bulk, and they are equally long. Then, the

left-lateral RS2 transitions into a passive state, gets abandoned and stops growing, while the right-lateral branch RS1 extends aseismically. We record no pronounced off-fault deformation for either RS1 nor RS2. They are rather localized fault strands that are wider than the newly formed faults in the other models, however.





In the rate-neutral model the propagating rupture excites several equidistant secondary splay faults that form under the same Coulomb-angle (R1) and saturate at x∼80 km, such that they are 16.5 km long as a maximum. As the main rupture and the secondary ruptures penetrate into the bulk, two of the secondary splay faults that were triggered by the dynamic main fault rupture continue to grow by ∼5 km at a constant Coulomb-angle, and then get abandoned. Simultaneously, the main rupture initiates RN1 and RN2 under the respective Coulomb-angles of the Riedel shear and the conjugate shear. Right after the earthquake, the left-lateral RN2 is ∼4 times longer than RN1. However, right-lateral RN1 is increasingly favored as both fault branches extend in an aseismic manner and RN1 finally becomes the main extension of the predefined fault. The aseismic phase in which RN1 extends by 130 km and grows towards the right boundary lasts for 1.2 years.

In the rate-weakening model RW the secondary splay faults form earlier than in model RN, and are more localized and partly non-equidistant. Particularly apparent is a secondary splay at x∼105 km that is highly localized and causes a stress shadow on its compressional side leading to a splay gap. It thus prevents an equidistant spacing of the secondary splays and changes the angles of the subsequent forming splays. We investigate this behavior in section 3.4. As the secondary splays propagate into the bulk at the fault tip they generate approximately 7 km of new fault surface under the conjugate angle of fault RW2 which they are flanking. Subsequently, the secondary faults and the conjugate RW2 merge. In contrast to models RS and RN, model RW reveals a very weakly localized and short incipient fault RW1 (1.8 km). This fault stops growing because the stresses on the compressional side of the compound of secondary branches and the sinistral conjugate RW2 are dominant and limit the extensional side stresses of short RW1. Thus, RW1 remains short and abandoned after the earthquake until the simulation stops, such that the sinistral RW2 constitutes the unique extension of the main fault. The simulation RW stops as branch RW2 approaches the lower y-boundary at a higher angle. Consequently, the maximum fault length $F_L$ is shorter than in the other models.

The off fault deformation pattern in the rate-transition model is composed of similar features as model RS. The two branches RT1 and RT2 form as the main fault rupture penetrates into the undeformed bulk. The fault evolution in model RT spans over several earthquake cycles (Fig. 5). Both branches RT1 and RT2 are well developed like in the model RN. However, on the long-term branch RT1 grows faster than RT2 during both aseismic and seismic fault growth stages. It is noticeable that the incipient faults in model RT are surrounded by a wide fan of plastic strain that is absent in the other reference models. This fan arises because both RT1 and RT2 host seismic ruptures in contrast to the newly evolving faults in the other reference models. Thus, a plastic fan like the fans next to the predefined fault can grow. Additionally, more strain is accumulated due to a slower fault growth rate (Fig. 3), which prolongates the model run time compared to the other models and facilitates the formation of distributed strain zones. Strikingly, the fan of fault RT1 is on the compressional side, which is due to the increased angle of RT1 relative to the predefined fault where the fan is on the extensional side. Hence, the angle between RT1 and $\sigma_1$ is smaller than 45°, which implies yielding on the compressional side of the fault. In contrast to the three other reference models, faults RT1 and RT2 exhibit notable changes in the absolute fault angle $\beta$. These bends are ascribed to the difference between lower-angle aseismic and higher-angle seismic fault growth, respectively (Preuss et al., 2019). The repeated sequences of seismic growth increase the overall angle of RT1 compared to RS1 and RN1. Nevertheless, the overall growth contribution in model RT is dominated by aseismic growth. This is well visible in figure 5 where the aseismic growth increments are opposed to the





seismic ones. Seismic fault growth is mainly limited to the first earthquake where the off-fault damage and the fault extensions RT1 and RT2 are created. In the subsequent evolution only marginal portions of fault extension are ascribed to coseismic

345  events. This seismic contribution is accumulated at the outer edges of the interseismically deformed regions (figure 5). The reasons and implications of these findings are discussed in section 4. Additionally, faults RT1 and RT2 interact with each other. Fault RT2 starts to bend towards RT1 at 360 years. This behavior is not recorded in the other reference models. Visible is also a seismically initiated connection between RT1 and RT2 at x=130 km that starts at 355 years.

In summary, the strike angle of the formed faults increases from rate-strengthening to rate-neutral to rate-weakening material.

350  The case of a transition from RS to RW describes a more complex case with intermittent aseismic and seismic growth, fault bends, fault interaction and additional off-fault deformation. In addition, the degree of new fault localization is highest in model RW and follows the order RW-RN-RT-RS.

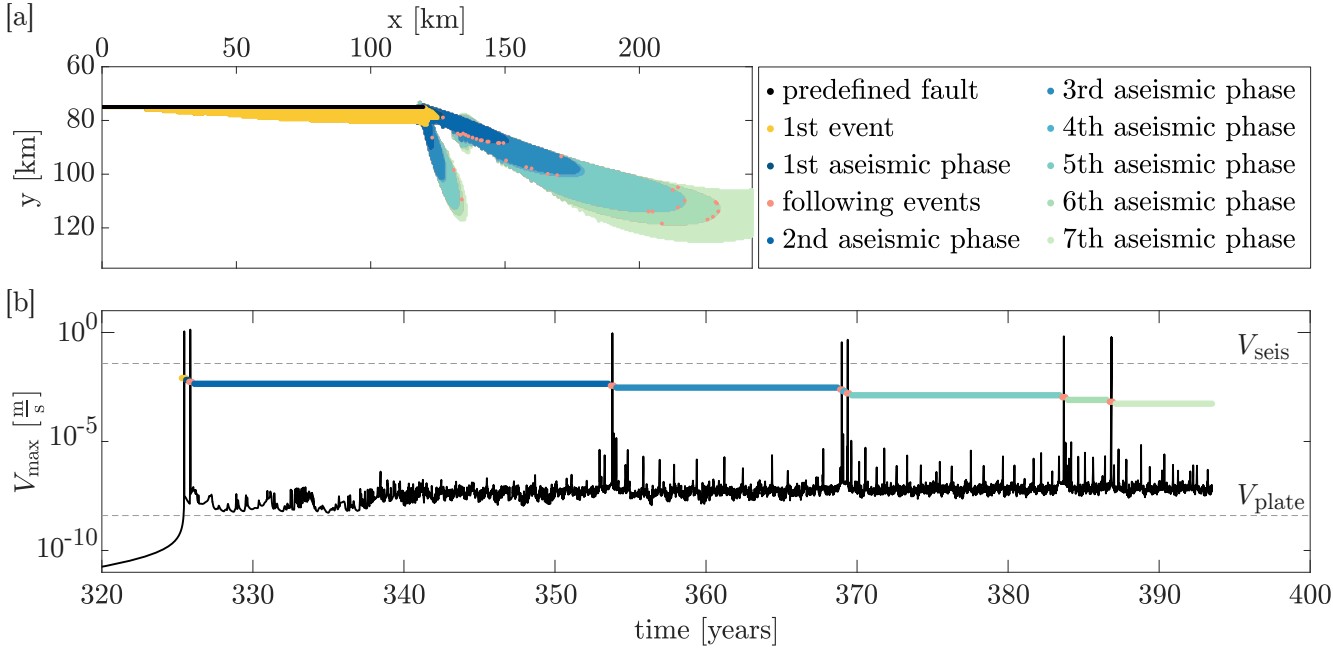

**Figure 5.** Temporal and spatial evolution of model RT with indication of aseismic and seismic fault growth stages in color. [a]: Accumulated plastic strain exceeding $\varepsilon_p > 4e\text{-}5$ for different growth stages. [b]: Logarithm of the global maximum slip velocity $\log(V_{\max})$. A video of the simulation can be found in the online material link to video model RT, 1.1 Gb

### 3.2  Role of a visco-elastic substrate on off-fault deformation

In this section we first analyze the properties of the saturating plastic off-fault fan and study the determining parameters for

355  this saturation. Secondly, we study the long-term effects of the relaxed visco-elastic substrate on the growth of the splay fault fan.





### 3.2.1 Off-fault fan and saturation

In all four reference models the plastic off-fault fan reaches a maximal width $H_{max}$ at distances over x∼80 km. The onset distance of the saturation does not depend on the bulk rheology. However, $H_{max}$ varies from 8.7 km in model RS to 8.2 km in model RT to 6.4 km in model RN to 6.2 km in model RW. In figure 6a we show that the slip during the first earthquake saturates, i.e. the rupture develops a flat slip profile. The reason for this behavior is the transition from a crack-like to a pulse-like rupture, which can be seen in the steep tapering off of the slip velocity (a healing front) in figure 6b when the rupture front reaches a distance x∼80 km. This transition to a steady pulse-like rupture results from the finite seismogenic depth. In 3-D rupture models, a healing front emerges when the rupture front reaches the bottom of the seismogenic zone, and then travels back to the surface, turning the initial crack-like rupture into a pulse. This interpretation of the saturation of the plastic zone thickness, based on dynamic fracture mechanics and 3-D simulations, was first proposed by Ampuero and Mao (2017). In our 2.5-D approach we do not model the actual rupture front nor the healing front at depth. However, the counteracting response of the visco-elastic substrate in the conservation equation of momentum (Eq. 2) captures the effect of stress transfer from depth to surface. The thickness $T_S$ of the lower crustal substrate can be set arbitrary large and is irrelevant for the model outcome.

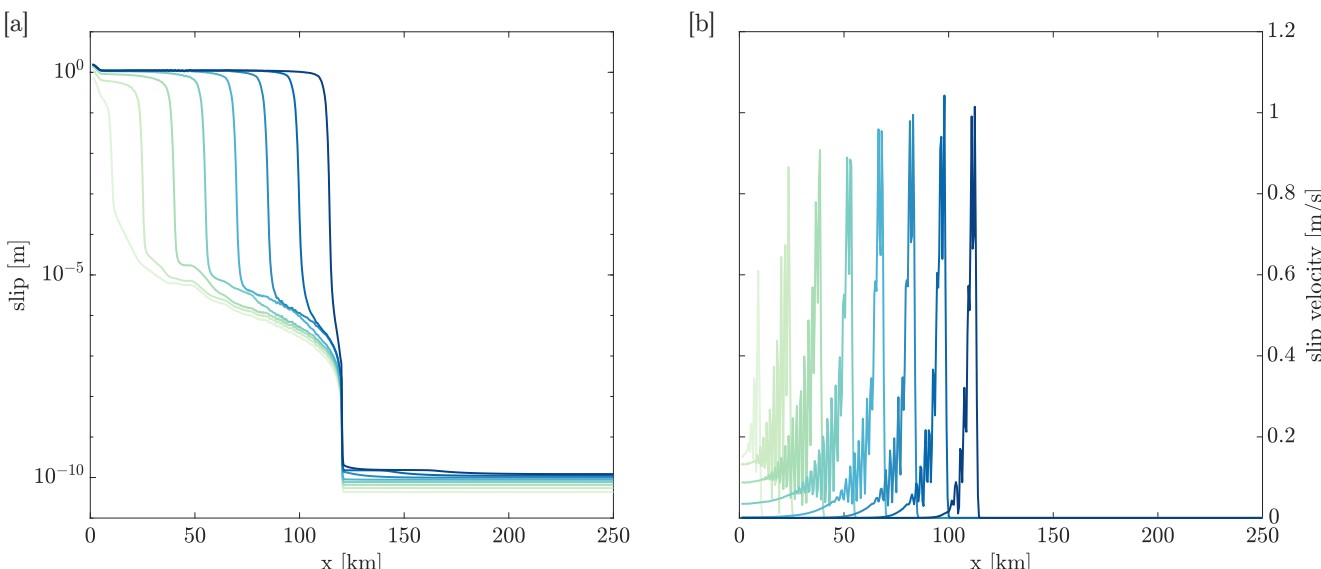

**Figure 6.** Accumulated plastic slip and slip velocity during the first earthquake on the main fault plotted in 5 s intervals.

We investigate the impact of crustal thickness $T$, initial background pressure $P$, initial bulk host rock state variable $\theta_{hr}$ and $(a-b)$ on the maximum plastic fan width $H_{max}$. In our simulation the fan width increases proportional to the crustal thickness $T$ (Fig. [7a]). The onset distance of the saturation $x_{H_{max}}$ is also proportional to $T$ (Fig. [7e]). Both characteristics are due to the 2.5-D effect of our simulation. In essence, this $T$-$H_{max}$-relation agrees with the simulation data of Ampuero and Mao (2017, inset of figure 13.5 therein). However, our simulation results coincide better with the theoretically determined linear relation between $H_{max}$ and $T$ at large $T$ than with the nonlinear trend in the 3-D simulation (Ampuero and Mao, 2017). This is due to a





small ratio (nucleation size)/(seismogenic depth) = 0.5 in our simulations, which favors a faster covergence of the curve to the linear trend. Furthermore, the different values of $H$ between both studies are due to different plastic strain cut-off levels used in the definition of $H$.

We additionally vary the initial background pressure $P$ from 10 MPa to 40 MPa in steps of 10 MPa and find that the
maximum fan width converges to $H_{\max} \sim 8.5$ km, which is 0.3 km wider than $H_{\max}$ of the reference model RT (Fig. [7b]). This implies that $H_{\max}$ is less sensitive to variations in lithostatic pressure with depth than to the crack-to-pulse-like rupture transition induced by the depth of the seismogenic zone. Our result is consistent with the theory of Ampuero and Mao (2017, their equation 13), who found that the maximum fan width is independent of normal stress. A reduction and increase of the initial host rock state variable $\theta_{\mathrm{hr}}$ reveals that the value corresponding to the reference model results in the greatest fan width,
while higher and lower state values lead to a reduction of $H_{\max}$ (Fig. [7c]). A change in $(a-b)$ from 0.001 to 0.01 shows that the fan width $H_{\max}$ converges to a value of 6.4 km for decreasing $(a-b)$, which corresponds to the fan width of the rate-neutral model in which bulk $(a-b) = 0$ (Fig. [7d]). However, since bulk $(a-b)$ is positive in model RT that we analyze here, no localization is reported in contrast to model RN or model RW. A maximum in the $(a-b)$-$H_{\max}$-relation is reached for $(a-b) = 0.006$ with $H_{\max} = 10$ km. If $(a-b)$ is increased above 0.006 the fan width drops to values below 4 km. This is
due to a factor 2 decrease of the slip velocity $V_{p_{\overline{\max}}}$ (Fig. [7f]). Although $V_{p_{\overline{\max}}}$ seems to be anti-correlated with $H_{\max}$, which in particular is observable for the crustal thickness and pressure model variations, it has an impact on the fan width in the case of higher $(a-b)$.

### 3.3 Role of the predefined fault angle on off-fault deformation

In sections 3.1 and 3.2.1 we demonstrated that the plastic strain produced during the first earthquake reaches similar values
for all four reference models. In the presence of localized deformation (models RN and RW) the splay faults form at a greater angle than the predefined main fault. Thus, the optimal angle of a newly forming fault in a dynamically elevated stress field does not coincide with the predefined fault strike. We suppose that the majority of the off-fault strain in these models is due to a 'misalignment' of the predefined fault together with the dynamic reorientation of stresses. In order to prove this supposition we design model OOF with an optimally oriented fault in which the predefined fault strike is at $15°$ to the horizontal shear
direction. This is the optimal strike angle, which we obtain by inserting the static friction coefficient $\mu_0 = 0.6$ in equation 16. The parameters of model OOF are based on model RT. In figure 8 we compare the amount of plastic off-fault energy between model RT and OOF. While the reference model RT reaches typical plastic energy values and a typical ratio of on-fault to dissipated off-fault plastic energy (e.g., similar to Okubo et al., 2019), these values for model OOF are substantially lower. In fact, model OOF exhibits almost no off-fault deformation (Fig. [8a]) and the amount of plastic energy dissipated to the
off-fault material is only 0.2 % (Fig. [8c]). This means that the different strike angle of model OOF reduces the magnitude of the off-fault stresses and thus the reach of off-fault deformation. The minor off-fault deformation starting at x $\approx$ 90 km is due to the dynamic reorientation of stresses, which leaves the pre-existing fault slightly misaligned with the dynamic stress field at high slip velocities.





**Figure 7.** Impact of various parameters on the maximum plastic fan width $H_{\max}$: [a]: Crustal thickness $T$; [b]: Initial background pressure $P$; [c]: Initial bulk host rock state variable $\theta_{\mathrm{hr}}$ and [d]: $(a-b)$. [e]: Relation between onset of the saturation $x_{H_{\max}}$ and $T$. [f]: Average slip velocity during the first earthquake $V_{p_{\max}}$ as a function of the highest and lowest parameter of each of the four model variations ($T$, $P$, $\theta_{\mathrm{hr}}$, $(a-b)$), respectively.

In figure [8b] we compare the dynamic stress drop between model RT (3.9 MPa) and model OOF ( 5.8 MPa). The difference
of 1.9 MPa reveals that more energy is concentrated on the predefined fault in model OOF. Additionally, the process of the
stress drop is 3.8 s faster than in model RT. We can note that this difference cannot be explained by the theoretical relation in
Ampuero and Mao (2017, equation 13), in which $H_{\max}/W$ is proportional to the ratio of stress drop to strength drop squared.
This points to opportunities to refine that theory. In summary, a fault at an optimal angle in the interseismic sense results in
a much smaller off-fault plastic yielding during earthquakes. However, due to dynamic reorientations of the fault-near stress
field minor off-fault yielding still occurs.







**Figure 8.** Comparison of plastic off-fault strain and energy between model RT and OOF. [a]: Accumulated plastic off-fault strain for both models in gray-scale during the first earthquake. Red triangles indicate the sample location of the stress data plotted in [b]. Panel [c] shows the ratio of the plastic energy opposed to the total energy in percentage. The plastic energy is subdivided in an on-fault (blue) and an off-fault (red) part.

## 3.4 Role of higher pressure and a thicker crust on secondary off-fault localization and main fault replacement

In order to increase the strong off-fault localization of reference model RW, with its propensity towards irregular fault pattern and unevenly spaced secondary fault branches, we increase the initial background pressure $P_B$ of model RW by a factor of 2 (to 40 MPa). Additionally, we increase the crustal thickness $T$ by a factor of 2, to 40 km, which is a typical value for continental

crust and furthermore enhances the extent of off-fault plasticity (section 3.2.1). This model is named HPT. The rate-weakening behavior of the bulk is kept as in model RW to facilitate fault localization. The primary earthquake in model HPT occurs after 656.3 years, which approximately corresponds to twice the event time in model RW. The time lag is caused by the doubled $P_B$





and the thicker crust. As the rupture propagates along the predefined main fault, secondary ruptures create localized secondary splay fault branches like in model RW. However, in contrast to model RW, the splays are sharply localized as soon as the

off-fault deformation occurs (Fig. [9b]). The main fault rupture induces four dynamically rupturing secondary Riedel splays HPT1, HPT01, HPT001 and HPT0001 under the Coulomb angle. In turn, these splays induce some tertiary ruptures. Each of the dynamically created incipient faults has an extensional and a compressional side, like the main fault. Interestingly, the secondary splays in model HPT do not saturate in contrast to model RW, in which an upper bound $H_{\mathrm{max}}$ exists. This is caused by the higher energies of the individual ruptures due to the higher initial background pressure. Hence, this is an indication that

the counteracting stresses at the base of the crust are too small to limit the extent of splay faulting in this setting. This behavior is particularly visible for the outermost branch HPT0001, which is most free to grow and barely interacts with other ruptures as it diverges from the predefined fault. Besides the four Riedel faults, model HPT additionally produces high-angle secondary conjugate faults. This happens even before the main fault rupture penetrates into the bulk at the tip of the predefined fault (fault strands HPT2, HPT02 and HPT002). Some of these conjugate faults extend to connect the main fault with the secondary splays

and then stop growing when they intersect with the next fault strand (Fig. [9c]). The resulting fault pattern has similarities with fault structures observed in nature and their schematic interpretations (compare to Fig. [1]). Another effect of higher $P_B$ and thicker $T$ is that the lateral spacing between the separate branches increases such that they become individual independent fault strands. Concurrently, these fault strands can interact with each other due to an interference of the individual local stress fields with the ones of the surrounding dynamic ruptures. This leads to irregularities in splay spacing, splay bending and abandoning

of splays. We follow that a higher background pressure increases the degree of localization of the splays and affects the spacing between the splays. Additionally, the complexity in the localized off-fault deformation is increased and the combination with a higher crustal thickness enhances the spatial reach of the off-fault splays. The latter is facilitated by the rate-weakening bulk.

In the following we analyze several indications of fault and rupture interactions due to stress changes that are typically ignored in seismic cycle models. They include: (1) Rupture arrest when two sub-parallel ruptures get too close to one another.

This can be observed for fault HPT001, which stops growing because the stresses on the extensional side of the subsequently forming branch HPT01 increase, get dominant and limit the compressional side stresses of HPT001. As a consequence, only extensional stresses remain at the tip of HPT001, such that the fault gets thinner on its compressional side (Fig. [9c,b]). This leads to (2) a stop in fault growth and fault abandoning. Further, fault bending (3) is observed as fault HPT02 approaches HPT0001 and the former starts to bend due to local interactions of stresses. After bending, both faults intersect (4), which causes

HPT02 to terminate (5). All together, this behavior is well visible in the video of the HPT simulation, 19 Mb. Consequently, new interjacent branches can stop if their extensional side stress field interacts with the compressional side stress field of another rupture. This is the case when the branches of two subparallel ruptures get close to one another. In this process, the fault on the extensional side is likely to continue extending. This line of reasoning applies for a dextral fault system and is reasonable since the evolving fault structure as a total has an extensional character, which means that an extensional stress state

is predominant and the extensional fault's side is favored.

The main fault rupture forms a Riedel fault HPT1 and a conjugate HPT2 like in model RW. These two faults and the secondary branch HPT01 grow until the event slip velocity drops to 8e-4 m/s after a duration of 330 s. Then, they stop growing

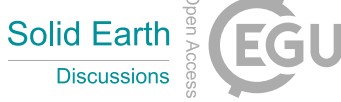



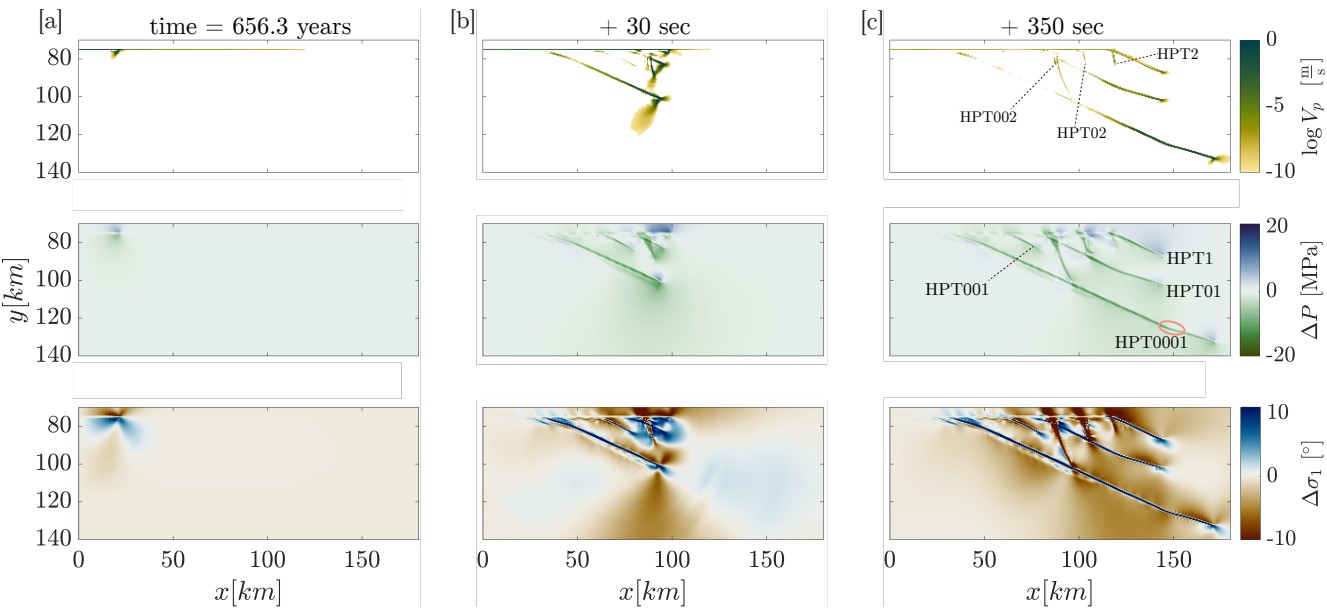

**Figure 9.** Model HPT with higher background pressure ($P_B$ = 40 MPa) and increased crustal thickness ($T$ = 40 km). Plotted are the logarithm of the slip velocity $V_p$, the pressure-difference $\Delta P$ between initial background pressure $P_B$ and dynamic pressure $P$ and the dynamic change in $\sigma_1$-orientation $\Delta \sigma_1$ for three different instants [a], [b], [c], respectively. Different fault branch names are indicated in [c]. Red circle in [c] marks the bend of branch HPT0001 due to an aseismic transition phase between two successive earthquakes.

and only the outermost splay HPT0001 continues to grow as the slip velocity again rises to the seismic range. The drop in slip velocity leads to a fault bend (red circle in Fig. [9c]). Since the outermost fault is the only active fault at the end of

the simulation we infer that during the evolution of the fault network HPT a main fault change occurred. Hence, the branch HPT0001 is most favorably oriented with the local and far-field stresses and replaces the predefined straight main fault. We refer to this dynamic process as a main fault replacement.

## 4   Discussion

Our results suggest that the fault growth process in nature is best represented by bulk behavior that transitions from rate-

strengthening to rate-weakening with plastic strain. Constant rate-dependent behavior in the bulk results in very fast fault extension with unrealistic rates of up to 77 km/yr. All four reference models agree to first order with the finding that the maximum amount a fault can grow in a single earthquake that ruptures the entire fault is of the order of 1% of its previous length (Cowie and Scholz, 1992). The distance a fault can grow in one event also depends on the optimality of the fault orientation and on the bulk rheology prior to the rupturing. In model RT, for example, only the first event extends the fault

significantly, while the subsequent events occur after the fault orientation has already adapted to the far-field stress and extend the fault only marginally (Fig. 5).





## 4.1 Riedel shear splays

The concept of work minimization states that new faulting starts when the active fault has become sub-optimal in the Coulomb sense, inefficient and sufficiently high amounts of strain are transferred into the surrounding rock (e.g. Cooke and Murphy, 2004; Cooke and Madden, 2014). In fact, natural faults are often unfavorably oriented with respect to remotely acting stresses (e.g., Faulkner et al., 2006). In this case new, secondary faults form at acute Coulomb angles to a primary fault (Scholz et al., 2010). Several studies linked Riedel shears and Coulomb shears (e.g., Tchalenko, 1970). In all our reference models a dynamically formed Riedel fault R1 and a conjugate R2 (R') emerge at the fault tip. In the rate-weakening and rate-neutral models, Riedel faults dynamically initiate off-fault and grow during the first earthquake, which indicates that the seed fault was not optimally oriented relative to the local dynamic stresses (Fig. 4, discussed in section 4.4). Interestingly, most of the dynamically generated Riedel faults get abandoned after they form. An exception is the Riedel fault at x=105 km in model RW, which was generated during the first event and grew aseismically 0.35 years later (see video RW, 170 Mb). It is an example of a fault that is excluded from the saturating effect of the lower crustal substrate (discussed in section 4.7). A potential reason might be that the counteracting stresses of the crustal substrate caused by the primary event have ceased and the substrate is again relaxed. This means that the saturation of off-fault deformation thickness does not apply to slow growth processes whose time scale is longer than the deep relaxation time scale as was first proposed by Ampuero and Mao (2017).

We analyzed the angles of newly formed Riedel faults R1 and R2 and showed that they comply with the Mohr-Coulomb faulting theory. Earthquakes on the main fault induce a dynamic elevation of the local stress and friction coefficient and a lobe-like alteration of the stress orientations. These dynamic changes determine, via classical failure theory, greater fault angles than the typical $10° - 20°$ range reported in experimental studies (e.g., Moore et al., 1989; Tchalenko, 1970). We reported this behavior already in a previous study (Preuss et al., 2019). Here, we additionally observe that the conjugate R2 responds to dynamic stresses with a decrease in $\beta_2$ due to its antithetic nature (sinistral fault in a dextral fault system). Hence, with respect to the absolute fault angle, the seismic contribution is contrary to that in a dextral fault like R1. The angle of R2 seems high, although it forms according to the classical faulting theory. A high angle was also reported in a computational study of dynamic rupture allowing for formation and growth of secondary faults during a single earthquake (Kame and Yamashita, 2003). Stress analysis of a crack loaded in mode II explains the formation of tensile fractures at the crack tip (King and Sammis, 1992).

## 4.2 Fault bending due to earthquakes

Owing to the differences between quasi-static and dynamic stress and strength conditions, the faults in our models reorient and bend as they alternate between aseismic and seismic growth stages. The fault angle $\beta$ changes after each earthquake in all models. This behavior is especially well visible at the $\sim 30°$ 'big bend' of R1 at the tip of the pre-existing fault at x = 120 km (Fig. 3). Here, during the first earthquake, the pre-existing fault is severely misaligned with the locally rotated stresses. Thus, as the rupture reaches the tip of the fault and penetrates into the bulk, the fault bends under a great angle. In the following aseismic stage $\beta$ decreases. This behavior leads to multiple smaller fault trace bends on the long-term in the case of model RT





with several earthquakes (Fig. 3). This history is well traceable in the increments of fault growth in figure 5. The bends become
less and less pronounced in the later stages of fault evolution because first, the contribution of seismic fault growth reduces
over time; and second, the fault system tends to optimize its growth efficiency and reaches a steady state in which seismic and
aseismic growth happen in the same direction (earthquakes 5,6,7 in Fig. 5). These two reasons are interconnected. On average,
the fault bends in model RT are in the order of $\sim$10° - 15° during fault formation, but the individual bend angles decrease
over time. These values fit well within the range of 6° - 20° reported by Moore and Byerlee (1991) and the average splay bend
angle of $\pm$17° (Ando et al., 2009) for the San Andreas Fault. Additionally, these values support the statement of Preuss et al.
(2019) whose stress field analysis of the Landers-Kickapoo fault suggests that an angle greater than $\sim$25° between two faults
indicates seismic fault growth.

To summarize, our findings imply that fault bending is most likely the result of a misalignment of the pre-existing fault, which
can occur also in a frictionally homogeneous medium. This fault misalignment can be strongly affected by seismically activated
dynamic processes. Fault bending must not necessarily be the result of only seismic rupturing but the magnitude of bending
can be strongly increased by it. Additionally, modeling shows that bending related to seismic rupture smears out over time but
an overall increase of the angle of the entire fault trace can be recorded on the long-term.

### 4.3 Contribution of aseismic and seismic fault growth

We show that seismic growth has a visible influence on the overall fault trace angle, which is reflected in the $\sim$14.2° greater
fault angle of the partly seismic and partly aseismic extending RT1 compared to the purely aseismic extending RS1 in the rate-
strengthening model (Fig. 3). However, in general fault growth in our model is mainly restricted to an aseismic contribution,
because a seismic rupture only reaches the current fault tip if this part of the fault is already highly localized. This is solely
the case in the first earthquake when the entire fault trace coincides with the predefined, weak, mature fault. In the subsequent
earthquakes, the rupture only breaks the fault at seismic slip rates far behind the current fault tip (to the west) and the seismic
rupture stops before reaching the actual fault tip (see video model RT, 1.1 Gb). Hence, the contribution of the seismic events
is rather limited to further localizing deformation in areas of initialized distributed yielding and to increasing the overall fault
trace angle than to extend the fault significantly at the tip.

### 4.4 On the optimality of the pre-existing fault angle

Our study shows that the amount of off-fault deformation is crucially dependent on the misalignment of the fault or, in other
words, on the optimality of the angle of a predefined fault (section 3.3). For a model with an optimally oriented fault in the
interseismic sense we report a 6.5-fold plastic energy decrease with respect to a fault that is parallel to the loading direction
(Fig. 8). This decrease in expended energy results in significantly less plastic deformation off the main fault. The same model
with an optimally oriented fault shows a 1.7-fold decrease of plastic deformation on the main fault and a stress drop increase
by 149% with respect to model RT. These numbers reveal that the initial orientation of a fault subject to dynamic earthquake
rupture with off-fault deformation is essential for the amount of off-fault deformation. Data from locked strike-slip faults in
California confirm that stress drops are larger on faults with a greater measured Riedel angle (Moore and Byerlee, 1992). An





equivalent to varying the initial fault angle is to vary the initial $\sigma_1$-direction in the simulation. A number of studies investigated the effect of the far-field stress direction on the off-fault deformation or on the angle of dynamically rupturing secondary

fault branches (Templeton and Rice, 2008; Kame et al., 2003; Rice et al., 2005; Bhat et al., 2007). However, these simplified studies (simple linear on-fault slip-weakening law and constant or linear slip-weakening off-fault law, simplified elastic or elastic-plastic off-fault rheology, predefined secondary fault paths, no possibility of re-nucleation of rupture and artificial event nucleation) omitted to quantify the amount of plastic off-fault strain energy related to different angles between fault and $\sigma_1$-direction.

545       Our findings regarding the time-dependent optimality of the fault angle have implications for nature and for future dynamic rupture modeling studies. Active fault strands in nature that are surrounded by severe localized or diffuse damage zones, possibly extending far into the host rock, are strongly misaligned with the interseismic far-field stress field. This misalignment may be increased dynamically during seismic rupturing. This means that individual fault traces may reflect the local geology, structure or stress state rather than the prevailing far-field, long-term stress field and this effect would vary from segment to

segment randomizing the fault pattern (Moore and Byerlee, 1989). This explains the complex nature of inter-branched crustal fault systems.

These statements are supported by model HPT, where strong local alterations of stresses lead to marked secondary rupturing and a main-fault replacement (Fig. 9). Moderate off-fault deformation close to a fault suggests that the fault is slightly misaligned in interseismic phases, which may again be amplified by dynamic reorganization of stresses. Absent or very little off-fault

yielding indicate that the respective fault is well aligned with the far-field long-term stress field and dynamic rupturing on the fault has a minor effect on the off-fault stresses. Further, we showed that a higher dynamic stress drop can be taken as evidence for a well-aligned fault trace. A higher angle of the secondary splays indicates a stronger dynamic elevation of local stresses and friction due to increased slip rates, which increases the misalignment of the pre-existing fault in a dynamic sense. This statement is supported by a comparison of stably sliding and stick-slip segments in laboratory fault zones and the San Andreas

fault (Moore and Byerlee, 1991). In nature it is very likely that faults adapt to the regional stress-field on the long-term (e.g., Nur et al., 1989; Swanson, 2006, 1992; Katz et al., 2004; Chester and Chester, 1998; Vermilye and Scholz, 1998).

It is noted that the typical dynamic rupture modeling setup of a $0°$ pre-existing fault in a constant stress field with $\sigma_1=45°$ represents a very particular case. This case seems well suited to study reactivation of formerly passive faults. However, it seems less well suited to study realistic faulting processes on the long-term that are interacting with earthquakes, which alter the local

stress field and friction values.

## 4.5   Interaction of fault branches to optimize the fault system efficiency

We found evidence for connecting fault segments, which highlights that different fault strands interact both during and between earthquakes. Dynamic interaction is versatile and pronounced in model HPT (Fig. 9 and video of the HPT simulation, 19 Mb). Instantaneous weakening of the local friction coefficient and an instantaneous local stress rotation at the tip of each

dynamic rupture leads to complex faulting behavior due to an interference of the individual local stress-fields with the ones of the surrounding dynamic ruptures (section 3.4). The single main fault rupture in this model excites 10 dynamic secondary





ruptures on the extensional side bulk that can arrest, bend, converge, intersect and get abandoned. This complexity is linked to variations of the normal stress during earthquake sequences, which affect the evolving fault pattern. That behavior highlights the importance to include a varying normal stress in earthquake cycle models instead of assuming a constant normal stress.

We explain the tendency that the extensional side fault of two sub-parallel faults is the favored with the extensional stress state in a dextral fault system subject to a $\sigma_1$-direction of 45°. Supposably, a fault network in a compressive stress field favors fault growth on the compressive side in a likewise manner.

An interesting feature in model HPT is the main fault replacement (or jump). This is reflected in the singular growth and slip activity of the outermost fault branch HPT0001 at the end of the simulation. In the dynamically altered stress field this

outermost fault branch is most favorably oriented. A main fault jump was reported in southern California where the San Gabriel fault was originally the main strand of the San Andreas fault, but was replaced at about 4 Ma (Moore and Byerlee, 1991). Faults that are unfavorably oriented for large amounts of slip will be replaced by progressively better oriented faults (Moore and Byerlee, 1989).

Fault branch interaction occurs also on the long-term when the stress fields of approaching fault strands start to interfere

(manifest in a seismically initiated incipient connection between RT1 and RT2 at x∼140 km in Fig. 3). Seemingly, the fault system intends to increase its efficiency by decreasing the fault complexity on the long-term due to fault interaction which can lead to abandoning of abundant fault strands. This is another indication, apart from the previously discussed one, that the fault optimizes its growth efficiency and aims at reaching a steady state on the long-term in which seismic and aseismic growth preferentially happen in the same direction.

**4.6 Wing crack transition and relation to normal and reverse faults in the Anatolian Fault system**

The decreasing strike angle of fault RT2 brings another aspect with it. The fault RT2 is initially formed at the typical angle of a conjugate Riedel fault (R') and agrees with the standard Mohr-Coulomb failure criterion (Fig. 4). The predictions during the earthquake agree very well with the angles determined in laboratory experiments (Tchalenko, 1970). In the subsequent aseismic stage the angle of the conjugate RT2 starts to decrease. This behavior leads to a bending of RT2 on the long-term (Fig. 3, video

RT, 1.1 Gb). The conjugate angle decreases from an initial value of 72° to an angle of 56° at the end of the simulation. A tension fracture or wing crack forms perpendicular to the orientation of the minimum principal stress direction (Dooley and Schreurs, 2012). In our 2.5-D plane strain model this implies a formation parallel to the $\sigma_1$-direction. One could argue that the fault RT2 is undergoing a transition phase from a conjugate Riedel fault to a tension fracture. Before this transition is completed the simulation is stopped because RT1 has reached the eastern model boundary. Two more characteristics amplify this argument.

Firstly, the counterclockwise bending behavior of RT2 corresponds to the one of a wing crack (=tension fracture) in laboratory experiments (illustrated in Fig. [1d], (e.g. Willemse and Pollard, 1998)). Secondly, we record a significant amount of opening for RT2, which varies in the range of 50% - 100% compared to the shear component of RT2. This is another similarity to a classical tension fracture, which is typically an opening mode crack (Willemse and Pollard, 1998). Nonetheless, the dominant shear component alludes to a transition or an approximation to a wing crack rather than the development of a classical wing

crack.





Such classical wing cracks are typically found in laboratory experiments and as subsidiary cracks in nature (e.g. Lee et al., 2016; Birren and Reber, 2019; Mutlu and Pollard, 2008; Willemse and Pollard, 1998). In the following we briefly attempt to link our model observations with the theory of wing cracks and normal and reverse faults in nature. Similarities between wing cracks and normal faults at a larger scale were reported before (e.g. Mutlu and Pollard, 2008). A modeling study of
the Anatolian Fault system linked mode I wing crack formation to large scale mode I failure representing dyke, normal fault or rift formation (Hubert-Ferrari et al., 2003). Their Coulomb shear failure map predicts left-lateral optimum fault formation in areas of increased tensile stresses compatible with the creation of the East Anatolian Fault. Furthermore, their associated left-lateral optimum faults coincide well with mapped normal faults branching off under steep angles from the main North Anatolian strike-slip fault (see supplementary material of Perrin et al., 2016b). Many of such normal faults and also branching
thrust faults are found in the Cinarcik Basin, the Central High area south of Büyükçekmece and the Central Basin south of Marmaraereğlisi (Le Pichon et al., 2003). A perfect example is the Gölcük normal fault just west of the Izmit fault that is oriented at 46° to the main North Anatolian fault trace (Barka et al., 2002) and also small-scale normal faults with similar trends and 45° oblique to the master fault (Alpar and Yaltirak, 2002). Thus, transtensional and normal faulting that are located near pull-apart basins in the North Anatolian fault system (Ickrath et al., 2015) are geometrically very similar to lab-scale
opening wing-cracks and the fault RT2 of our simulation. They evolve in a tension-gash just like wing-cracks (Sengör et al., 2004) and RT2 develops in the direction of such a tension-gash. Additionally, their transtensional nature is reflected in the shear- and opening-component of RT2, which is a necessary prerequisite for extensional faulting. Another great example of normal and thrust faults bordering an advancing strike-slip fault is the Altyn Tagh fault (see map in Perrin et al., 2016b). Some of the branching splays in this example are marked by the same bending behavior as RT2 farther away from the main fault.

### 625   4.7    Width of the off-fault fan

In section 3.2.1 we showed that the implementation of the 2.5-D approximation has a limiting effect on the width of the dynamically created plastic fan off the main fault, which is observed for the width of the inner damage zone in 3-D numerical models and in nature (Ampuero and Mao, 2017; Savage and Brodsky, 2011; Perrin et al., 2016b). This limiting effect is controlled by the transition from a crack-like to a pulse-like rupture shown in figure 6, which is evoked by counteracting
stresses in response to the sudden excitement of earthquake forces at the surface of the relaxed lower crustal substrate. The width of the fan is controlled by several parameters, of which the thickness of the elastic layer $T$ on top of the visco-elastic half-space has the greatest impact. Too high values of $(a - b)$ and too high and too low initial bulk host rock state variable values $\theta_{\mathrm{hr}}$ decrease the fan width significantly. If $(a - b)$ decreases towards 0, the fan width approaches the fan width of the rate-neutral model for which $(a - b) = 0$. A change of slip velocity due to the variation of one parameter does not affect the
plastic-fan width (Fig. 7). Another effect of the visco-elastic lower crustal substrate is the delayed onset of the first dynamic earthquake due to the viscous contribution of the lower layer compared to a pure 2-D simulation in which the crust is infinitely thick.





## 4.8 Modeling limitations and future work

The natural faulting process a three dimensional process. Compared to previous studies that applied the STM-code we here
approach three-dimensionality by a 2.5-D approximation. We thus obtain a finiteness of the seismogenic depth that limits the
stress concentration at the fault tip, which in turn limits the spatial extent of plasticity outside the main fault (Ampuero and Mao,
2017). However, this approximation assumes a simple linearly elastic crust and computes the thickness averaged stresses in it
due to traveling rupture zones. This approximation does not actually account for the third dimension and neglects parameter
variations with depth as well as a possible change of the fault dip angle with depth. In this study faults are always vertical in
a plane-strain sense, cutting through the entire upper crustal layer. Further, the simulations exclude a temperature-dependent
rheology that would imply rheology changes with depth. The presented comparison of rate-strengthening, rate-neutral, rate-
weakening and a transition case between them can be seen as an insightful improvement compared to our previous study in
which we only uses a rate-weakening bulk material (Preuss et al., 2019). In particular it represents an improvement because it
simulates behavior observed in the laboratory. Here, we presented changes of frictional parameters $L$ and $(a - b)$ with plastic
strain. Additionally, we run test models in which we weakened $a$ instead of $b$ keeping the overall $(a - b)$ constant. Further, we
tested a simultaneous weakening of $a$ and $b$ keeping $(a - b)$ again constant. These different weakening scenarios do not change
the behavior of the model. However, changes of other frictional parameters or material parameters (e.g., shear modulus) with
plastic strain are not taken into account in our simulations despite they can be expected in natural fault systems. Our model is
a simplification in that it ignores anisotropy, poroelasticity, or dilatant volume changes, which are typically observed in natural
faults (e.g., Woodcock et al., 2007; Brace et al., 1966; Peacock and Sanderson, 1992; Rawling et al., 2002). In our previous
work we discussed the need for an alternative invariant continuum-based rate- and state-dependent friction formulation for
fault width $W$. The result of using the here proposed slip-velocity-dependent heuristic fault-width formulation is that both
the fault angle and the temporal onset of the earthquake converge with grid size at a resolution of 250 m (appendix A). Due
to computational time reasons all our reference models had to be run with a resolution of 500 m, however. With respect to
the note in section 4.4 of Preuss et al. (2019) we here want to add that our proposed heuristic fix needs further research
including the comparison to analog models to test and further refine the continuum-based constitutive relationship describing
self-consistently both localization toward a fault and deformation within the fault.

## 5 Conclusions

In this study we simulated the spatio-temporal evolution of a complex strike-slip fault system subjected to repeated earthquake
ruptures. We applied an invariant rate- and state-dependent friction formulation framework that allows for the spontaneous
growth and evolution of a fault. This STM-RSF framework was extended with a 2.5-D approximation, a new dynamically
adapting slip-velocity-dependent fault-width formulation and a plastic-strain-weakening mechanism of bulk parameters in-
spired by laboratory experiments. With this advanced model we present different possibilities of how a strike-slip fault grows
due to (a)seismic processes in different host rock rheologies of which the end-member cases are bulk-velocity-weakening and
bulk-velocity-strengthening. This work focuses on three main aspects: 1) It discriminates between the conditions leading to



distributed or localized dynamic off-fault deformation and the saturation of the plastic zone width. Our models distinguish between off-fault deformation geometries observed in nature (Fig. 1). 2) This study analyzes distinct extension styles of the main fault leading to a complex interactive fault network with bends caused by differences in angle between seismic and aseismic segments. The different fault branches are successfully linked to the Mohr-Coulomb faulting criterion. The development of

Riedel shear faults and their conjugates is caused by dynamic stress field effects and also explained by the theoretical faulting criterion. 3) Ultimately, our study demonstrates that the amount of plastic off-fault deformation crucially depends on both the initial fault orientation with respect to the far-field stresses but also on the dynamic optimality of the fault angle in relation to local stresses. The optimality of fault alignment in a stress field is time-dependent and depends on local variations of rotating stress orientations.

Additionally, we found that under the wide range of conditions explored the contribution of seismic fault growth is limited compared to the aseismic contribution. Earthquakes are rather leading to a greater localization in areas of distributed deformation close to the fault tip. Nevertheless, the overall fault angle of a fault that extends by combined aseismic and seismic growth is 14.2° greater than the fault angle of a purely aseismically growing fault. Further, the earthquakes evoke small segment bends in the order of ∼10° - 15° along a fault trace. However, to some extent these bends get smeared out over time as the fault

straightens gradually.

With respect to fault branch and rupture interactions we reported rupture arrest, fault bending, fault convergence and intersection, arrest of fault growth and fault strand abandoning. Fault interaction was observed on the long-term and during coseismic events. In an extensional fault setting the extensional side fault of two sub-parallel faults is the favored one and likely to continue. The dynamic rotation of stresses can lead to a reorientation of stresses, which might result in the severe misalignment of

the former main fault. This will lead to a replacement of the main fault trace and a jump of fault activity. Thus, fault systems tend to optimize their efficiency by adapting to changing conditions. We additionally found that fault systems optimize their growth efficiency by progressively favoring similar growth directions for seismic and aseismic growth.

With our work we provide the basis for simulations and analyses of complex evolving fault networks subject to long-term and short-term dynamics. The approach we presented has potential to be applied to a more realistic fault map in a future study.

*Code and data availability.* The repository cited in the references (Preuss et al., 2020) contains an executable, with which the reference model can be rerun. Figures 3 - 6 of this paper can thus be reproduced. Pending further notice the following link will lead to a temporary repository, in case the permanent repository is not yet accessible: temporary repository.

*Video supplement.* The repository cited in the references (Preuss et al., 2020) contains five videos showing the temporal evolution of the four reference models and model HPT, which are all discussed in the main text. Pending further notice the following link will lead to a temporary

repository including the five videos, in case the permanent repository is not yet accessible: temporary repository.





**Appendix A: Test of the slip-velocity-dependent fault-width formulation**

We tested the new slip-velocity-dependent fault-width formulation (equation 13) using 4 different resolutions ($\Delta x$ = 125 m, 250 m, 500 m and 1000 m). The used model setup is the one from Preuss et al. (2019) because in it faults are completely free to start growing from the center of the model without any pre-defined fault line but a small elliptical defect. In this model, faults grow both aseismically and seismically, however, under a different angle. This is considered in the convergence analysis, which shows that both seismic and aseismic fault angles converge with grid size (Fig. [A1 a,b]). Also, the time of the earthquake converges with grid size in similar manner (Fig. [A1 d]). We note that the model with the lowest resolution has a temporal onset of the event, which is only 0.86 years off from the highest resolved model. In conclusion and based on the convergence analysis of the new slip-velocity-dependent fault-width formulation, the authors choose to apply the second finest resolution, $\Delta x = 250$ m, for all model runs. This decision is mainly based on the fact that the highest variation is found in the seismic fault angles, while the seismic relative fault angle $\alpha$ as well as the seismic absolute fault angle $\beta$ converge for $\Delta x \leq 250$ m. The aseismic fault angles and the temporal onset of the earthquake start to converge before, i.e. for $\Delta x \geq 250$ m.

*Author contributions.* Simon Preuss designed the study, implemented and tested the 2.5-D approximation and the dynamically adaptive measure of fault width, designed the model setup, decided the parameter space, ran the simulations, gathered and interpreted the results, wrote the manuscript and organized the submission process. Jean Paul Ampuero provided the initial idea to use the 2.5-D generalized Elsasser approach, helped clarifying the results and improved the manuscript. Taras Gerya provided the initial idea and tests of the dynamically adaptive measure of fault width, assisted in formulating the 2.5-D approximation for STM-RSF and helped to design the initial model setup. Ylona van Dinther helped clarifying the results and improved the manuscript.

*Competing interests.* There are no competing interests.

*Acknowledgements.* The repository cited in the references (Preuss et al., 2020) contains four executables, with which the four reference models can be rerun. Figures 3 - 6 of this paper can thus be reproduced. Pending further notice the following link will lead to a temporary repository, in case the permanent repository is not yet accessible: temporary repository. This project acknowledges support through the SNF research grant 200021_182069. J.P.A. was supported by the French National Research Agency (ANR) through project FAULTS_R_GEMS (Grant ANR-17-CE31-0008) and Investments-in-the-Future project UCAJEDI (Grant ANR-15-IDEX-01). We thank Robert Herrendörfer for providing the STM-RSF code. We additionally thank André Niemeijer for a discussion on rate-and-state friction parameters for the bulk and Jianye Chen for giving very useful comments on the slip-velocity-dependent fault-width formulation. For constructive comments and discussions we thank the STM-group at ETH Zurich. Numerical simulations were performed on ETH clusters Leonhard and Euler. Perceptually uniform color maps are used in this study to prevent visual distortion of the data (Crameri, 2018).





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

990




**Figure A1.** Convergence analysis of fault angle and earthquake time as relation between grid resolution and fault angles and grid resolution and earthquake time start, respectively. Analysis is showing that all seismic and aseismic fault angles as well as earthquake timing converge with grid size. Color-filled symbols indicate different faulting stages. Blue errorbars correspond to errors in measuring the absolute angle $\beta$ and to spatial variations of the $\sigma_1$-direction at the fault tip (details are explained in Preuss et al. (2019)). The solid line corresponds to the standard deviation $[-1\sigma, 1\sigma]$, the dashed line to the standard deviation $[-2\sigma, 2\sigma]$. [a]: Grid resolution in meter versus relative fault angle $\alpha = \sigma_1 - \beta$ (see schematic illustration in [c]). [b]: Grid resolution in meter versus absolute fault angle $\beta$. [c]: Fault angle legend and schematic illustration of fault angles. [d]: Grid resolution in meter versus temporal onset of the earthquake in years.