# Peer review of "Characteristics of earthquake ruptures and dynamic off-fault deformation on propagating faults"

_Solid Earth, 2020_

## Referee Comment (RC1) · Michele Cooke (Referee) · 28 Mar 2020

This paper uses the Seismo-Thermal-Mechanical code with rate and state friction (STM_RSF) to explore lateral strike-slip fault propagation across multiple earthquake cycles. The work builds upon Preuss et al (2019) with additional formulations that allow for more realistic rheology. The findings present insights that until now were not available because the code used here can simulate the evolution of stress state throughout the both the aseismic and seismic portions of the earthquake cycle. The results of the parametric study presented here shows several interesting trends that warrant further investigation and could greatly impact our understanding of fault propagation over mul-

tiple earthquake cycles. I greatly appreciate the systematic nature of the presentation where the authors first test different rheology, then successively add new implementations to the code. The new implementation of fault width overcomes many disadvantages over other techniques and thus, provides a valuable approach for many future investigations. I offer a few suggestions that might strengthen the manuscript.

A. The assumption of lateral propagation of strike-slip faults does not consider how crustal faults might form by the upward propagation and/or linkage of early fault segments. Experiments of strike-slip fault evolution show upward propagation with the formation of an early set of echelon faults that link to form a through-going strike-slip fault (e.g., Tchalenko, 1970; Hatem et al., JSG 2017). We don't have reason to believe that crustal strike-slip faults would initiate differently from experiment observations. The text uses results of Perrin et al. (2016) that faults are most mature along their centers to justify lateral propagation. Lateral variation in fault maturity don't preclude early upward propagation that would produce echelon segments that may link earlier along some portions of the fault than others. Unlike the quasi-2D simulations in this paper, the base of 3D laboratory experiments distributes shear within the suprajacent material in a manner analogous to crustal systems where mid crustal deformation drives upper crustal faulting. I'm not saying that the investigation of lateral propagation of strike-slip faults within this paper is unreasonable. This is a great first step towards understanding the complex evolution of strike-slip faults but may be only part of the story. To strengthen the implications of the paper, the introduction and discussion of the manuscript should include consideration of the 3D context of these strike-slip faults. How might the findings differ if strike-slip faults initiate with upward propagation followed by linkage?

B. The conclusions that the strike-slip faults grow predominantly in the aseismic period of the earthquake cycle is based on the assessment that the rate weakening results, with their fast coseismic growth, are unreasonable. We do have evidence that faults can link during earthquake rupture along previously unmapped segments. This process of

quick coseismic linkage could look a lot like the results of the rate weakening models that propagate towards the edges of the model. If the models had a second fault segment, perhaps the rate weakening models would showing linkage of the segments in a very reasonable way. For this reason, perhaps the rate weakening rheology may be overlooked should not be considered completely unreasonable.

C. This question is related to the comment of B. Rate neutral and rate weakening are rheologies that we expect in the crust. In this study, they are excluded because the models do not produce more than one earthquake. What model parameters could be altered (for example, in a future study) to get more than one earthquake cycle for the rate neutral and rate weakening models?

D. Some parameters seem a bit outside of expected crustal ranges. The width of the plastic yielding fan seems large. The static frictional strength of the faults of 0.6 is high when we consider that crustal faults have fluids. The 20 km choice for maximum fault zone width for the heuristic fault zone thickness, needs stronger substantiation.

E. Something to consider in the discussion of the paper is the role of nearby faults on the 'bending' of fault traces. While fault strike may bend in response to changing slip conditions, most crustal faults develop within a complex system where they might interact with nearby faults. Slip on a nearby fault (such as the Garlock fault near the San Andreas) may in many cases have larger impact on the bending of faults than difference in aseismic and coseismic lateral propagation. Additionally, immature faults may develop bends in their earliest stages when neighboring segments that are not co-linear link up to form a single fault surface (Hatem et al., JSG 2017).

F. The manuscript strives to address a wide range of conditions/questions. I wonder if some parts of the manuscript, such as the HPT models, might be best served as supplemental material.

Specific comments

The paper is very well written. I have a few specific comments that may strengthen the writing in places.

Throughout the manuscript (eg. Line 102, 298 and many others): 'Fault extension' reads a bit odd since extension is a strain term. The text might be clearer with use of 'fault propagation'.

Line 40 (and there abouts): The use of Riedel terminology for splay fractures strikes me as a bit odd because we typically refer to Riedels as the early formed echelon fault segments. The fractures within the damage zone are more commonly called splay cracks. You may find papers by Cooke (JGR 1997) and Willemse and Pollard (1998) helpful because they show the range of orientation of splay fractures that can develop with different conditions on the fault.

Line 51: Define SCEC.

Line 213: Mixing strain (extensional) and stress (compressional) term. Make these both either strain or stress.

Line 252: spelling of strengthening

Line 253: 'This results' is ambiguous. This what? Being more clear will help the reader.

Line 254: Better than what?

Line 399 and throughout: I'm not a fan of the acronym OOF for optimally oriented fault model. Why not just spell it out since you already have a lot of acronyms and only use OOF for one section of the paper?

Line 497: This is just one paper, for which there is a rich literature. Add e.g. and some more citations.

Line 540: Another interesting study is a quasi-static dynamic model of Savage and Cooke ( JSG 2010). That study differs from the ones cited in that it does not limit damage development along pre-existing mesh. So, the results of Savage and Cooke

(JSG 2010) might be interesting to compare to your new model results.

Line 573-574: This is an interesting result. I believe that this finding confirms results of Jiang and Lapusta – it could be helpful to cite their work here.

Line 583: There is a rich literature on the development of new faults that are more efficient. Add e.g to this reference.

Line 594: Cooke (JGR 1997) show that changes in friction distribution near fault tips alters the stress concentration and the angle of the splay crack. The change in friction arises in the transition between mature fault with static friction to immature fault with higher friction. Could this process be contributing to your observation of changing splay angle?

Line 602: Add Cooke (JGR 1997) to this reference list as it is very much related to these other good papers.

Line 628: The width of the plastic fan in the models is larger than that seen in Savage and Brodsky (2011).

---

## Referee Comment (RC2) · Boris Kaus (Referee) · 30 Apr 2020

**Review of "Characteristics of earthquake ruptures and dynamic off-fault deformation on propagating faults." by _Preuss et al._**

_Boris Kaus, Mainz_

This is a very interesting and extremely well-written manuscript that employs a novel numerical approach that includes both long-term tectonic processes and short-term earthquake rupture to study off-fault deformation develops. A 2.5D approach is employed in which the effect of a viscous substratum is taken into account as well. Benchmarks are presented that demonstrate the robustness of the results w.r.t. numerical resolution, particularly for a newly proposed dynamic fault-width algorithm. Simulations are carefully performed and clearly described in which the complexity of the models is incrementally varied, such that readers get a clear understanding of the relative importance of the effects. The simulations reveal that there are quite some differences in model behavior depending on whether a rate weakening or rate strengthening rheology is employed. The most realistic results in terms of having successive earthquakes, the development of localized fault growth as well as distributed deformation is obtained with a rheology that transitions from rate strengthening to rate weakening. The implications of the models are discussed, as are the limitations and a potential application to the Anatolian fault system.
I believe that this is a very nice contribution that should certainly be published.

As you will see below, I have two moderate comments and a few minor remarks. I believe they would further contribute to the clarity of the manuscript and hope that the authors are willing to address them during the revision stage.

**1) Moderate comments**

**_Yield stress criteria_**
You correctly write that at yield $F=0$, and you employ a standard Drucker-Prager yield function (eq. 9). Yet, your expression for the yield function (eq. 8) is incorrect, which can be best illustrated graphically:

[Figure]

The plot shows the yield function (black) together with the Mohr-Circle (green circle, which has radius $\tau_{II}$). At yielding ($F$=0), the Mohr-Circle exactly touches the yield stress function. This condition is *not* $\tau_{II} = \sigma_{yield}$ (your eq. 4), as this gives the red circle (which predicts a stress that is somewhat larger than the yield stress). Instead, we can use trigonometry to compute the condition for $F$=0. If we define the effective angle of friction $\theta$ as:

$$\tan\theta = \mu_l(1 - \lambda),$$

we can define the yield condition ($F$=0) as:

$$\tau_{II} = P\sin(\theta) + C\cos(\theta)$$

Since this is correctly described in the textbook of one of the co-authors of this manuscript, I suspect that it is incorporated correctly in the software. I also don't know how big an effect it will make on the results, even if this would not be the case (to be tested). Yet, in any case, it would be good if you can correct your description.

**Elastic material parameters**
Your choice of having the same values for bulk and shear moduli (table 1; both 50 GPa) results in a Poisson ratio of 0.125. That might be appropriate for already damaged rocks, but perhaps not so much for intact rocks. How sensitive are your results to the particular choice of Poisson ratio?

2) **Minor remarks:**
- Table 1: I suppose that the host rock cohesion is 6 MPa, and not 6e6 MPa?
- Fig. 3: you show an overview of several different models. Yet, are the snapshots chosen to have approximately the same plastic strain, deformation stage or time? Would be good to mention it.
- Figure 4. It would be good to explain at the beginning of the figure caption that this figure concerns the RT model.

- Your movies are extremely large (some over 1 Gb!); it is certainly possible to create smaller movie-sizes from a set of pictures, and I believe that this is important for readers that do not have a high bandwidth connection. I personally use ffmpeg for this (free software for linux/mac). The following command creates a relatively modest-sized *.mp4 movie, if you have a series of pictures (here *.png named as MovieFrame.0001.png, MovieFrame.0002.png etc.):

```
ffmpeg -y -f image2 -framerate 10 -i MovieFrame.%04d.png  -vf pad="width=ceil(iw/2)*2:height=ceil(ih/2)*2" -f mp4 -vcodec libx264 -pix_fmt yuv420p Movie_loRes.mp4
```

---

## Author Comment (AC3) · 17 Jun 2020

17th June 2020
Dear Editor,
Thank you for your helpful comments on our manuscript entitled "Characteristics of earthquake ruptures and dynamic off-fault deformation on propagating faults" [Paper se-2020-16]. We are grateful for the constructive and thoughtful comments made by the reviewers. We have addressed their questions, which are quoted below in blue. Text in red indicates text added to the new version of the manuscript. We also provide a PDF version of the revised manuscript in which we highlighted the changes in red

(deleted) and blue (added). All line numbers in the letter below refer to the tracked-changes document. The equation numbering in this rebuttal letter does not coincide with the equation numbering in the original manuscript and tracked-changes document. We hope that our revised manuscript has clarified the questions raised by the reviewers and made the paper stronger.

Best regards,
Simon Preuss (on behalf of all co-authors)

_______________________________________________________________________-

Reviewer 2 - Boris Kaus

1.1) Yield stress criteria
You correctly write that at yield $F = 0$, and you employ a standard Drucker-Prager yield function (eq. 9). Yet, your expression for the yield function (eq. 8) is incorrect, which can be best illustrated graphically:

Fig. 1

The plot shows the yield function (black) together with the Mohr-Circle (green circle, which has radius $\tau_{II}$). At yielding ($F = 0$), the Mohr-Circle exactly touches the yield stress function. This condition is not $\tau_{II} = \sigma_{yield}$ (your eq. 4), as this gives the red circle (which predicts a stress that is somewhat larger than the yield stress). Instead, we can use trigonometry to compute the condition for $F = 0$. If we define the effective angle of friction $\theta$ as:
$\tan\theta = \mu_l(1 - \lambda)$,
Ăă we can define the yield condition ($F = 0$) as: $\tau_{II} = P\sin(\theta) + C\cos(\theta)$
Since this is correctly described in the textbook of one of the co-authors of this manuscript, I suspect that it is incorporated correctly in the software. I also don't know

how big an effect it will make on the results, even if this would not be the case (to be tested). Yet, in any case, it would be good if you can correct your description.

We thank the reviewer for this important comment and a clarifying discussion, which together helped improving the paper. We clarified the yielding function description and changed notations to avoid confusion. We use a modified Drucker-Prager yielding condition with constant compressive strength and variable friction coefficient. We changed the method description in line 141:

The onset of plastic deformation is defined by the yield criterion:

$$F = \tau_{II} - \sigma_c - \mu_{\mathrm{eff}}(\mathrm{RSF})\,P_{\mathrm{eff}}, \tag{1}$$

where $P_{\mathrm{eff}} = P - P_{\mathrm{fluid}} = P(1 - \lambda)$ with the pore fluid pressure factor $\lambda = P/P_{\mathrm{fluid}}$, $\sigma_c$ is the constant compressive strength that marks the residual strength at $P = 0$ and $\mu_{\mathrm{eff}}(\mathrm{RSF})$ is a variable effective friction parameter that we define based on our continuum RSF formulation. We use a modified Drucker-Prager plastic yield function (Drucker and Prager, 1952) in the form:

$$\sigma_{\mathrm{yield}} = C(\mathrm{RSF}) + \mu(\mathrm{RSF})\,P_{\mathrm{eff}},$$

where

$\mu(\mathrm{RSF}) = \tan(\sin^{-1}(\mu_{\mathrm{eff}}(\mathrm{RSF})))$ is the local friction coefficient that is widely used and obtained from laboratory experiments

$$\tag{3}$$

and

$$C(\mathrm{RSF}) = \sigma_c/\cos(\sin^{-1}(\mu_{\mathrm{eff}}(\mathrm{RSF}))) \text{ is the local cohesion.} \tag{4}$$

The local effective friction parameter $\mu_{\mathrm{eff}}(\mathrm{RSF})$ evolves according to the invariant reformulation of rate- and state-dependent friction for a continuum, introduced by Herrendörfer et al. (2018). This formalism was applied to freely and spontaneously growing seismic and aseismic faults by Preuss et al. (2019), by interpreting how plastic deformation starts to localize and forms a shear band that approximates a fault zone

of finite width that can host earthquakes. Localized bulk deformation and fault slip are related by defining the plastic slip rate $V_{\mathrm{p}}$ as

$$V_{\mathrm{p}} = 2\dot{\varepsilon}'_{II(p)}W, \tag{5}$$

where $W$ denotes the width of the fault zone in the continuous host rock. We formulate $\mu_{\mathrm{eff}}(\mathrm{RSF})$ as:

$$\mu_{\mathrm{eff}}(\mathrm{RSF}) = a\ \mathrm{arcsinh}\left[\frac{V_{\mathrm{p}}}{2V_0}\exp\left(\frac{\mu_0 + \frac{C}{P} + b\ln\frac{\theta V_0}{L}}{a}\right)\right], \tag{6}$$

where $a$ and $b$ are laboratory-based, empirical RSF parameters that quantify a direct effect and an evolution effect of friction, respectively, $L$ is the RSF characteristic slip distance, $\mu_0$ is a reference friction coefficient at a reference slip velocity $V_0$ (Lapusta and Barbot, 2012), and $C$ is the cohesion as part of the state variable $\theta$ (Marone et al., 1992) that evolves according to the aging law:

$$\frac{d\theta}{dt} = 1 - \frac{V_{\mathrm{p}}\theta}{L}. \tag{7}$$

Additionally, we updated Figure 4 according to the recomputed relative fault angles based on equation 9.

1.2) Elastic material parameters
Your choice of having the same values for bulk and shear moduli (table 1; both 50 GPa) results in a Poisson ratio of 0.125. That might be appropriate for already damaged rocks, but perhaps not so much for intact rocks. How sensitive are your results to the particular choice of Poisson ratio?
We thank the reviewer for this question. A Poisson's ratio of 0.125 is on the lower end of values for rocks, but still common for a wide range of rocks as for example
shown in *Poisson's ratio values for rocks* (H. Gercek, 2007). Furthermore, we tested a range of shear moduli resulting in varying Poisson's ratios and found only a marginal impact on the model results. In particular, the main messages of our manuscript are not influenced by changes in the Poisson's ratio. We illustrate this by comparing the snapshots of simulations with Poisson's ratio of 0.125 and 0.25. We focus on the dynamically generated off-fault yielding at approximately the same deformation stage just before the rupture hits the end of the predefined fault. Both snapshots are attached. The differences comprise:

- First earthquake nucleates 75 years later for $\nu$ = 0.25.

- The maximum slip velocity is $\sim$ 0.08 m/s higher for $\nu$ = 0.25.

- The off-fault splay localization is more irregular with a higher degree of localization and a slightly higher off-fault reach for $\nu$ = 0.25.

Fig. 2

Fig. 3

In light of our answer above we add to line 703 of the manuscript:
Our choice of parameters results in a Poisson ratio of 0.125. Such a relatively low Poisson's ratio is on the lower end of values for rocks, but still common for a wide range of rock types as for example shown in Gercek (2007). To illustrate the impact of different Poisson's ratios we have tested a range of different shear moduli resulting in varying Poisson's ratios. These tests have shown that the main messages of our manuscript are not influenced by changes in the Poisson's ratio.

2) Minor remarks:

Table 1: I suppose that the host rock cohesion is 6 MPa, and not 6e6 MPa?

This is correct. We changed that.

Fig. 3: you show an overview of several different models. Yet, are the snapshots chosen to have approximately the same plastic strain, deformation stage or time? Would be good to mention it.

Thanks for this comment. We added the following to the figure caption:

The snapshots in [a] are chosen to have approximately the same deformation stage with regard to fault length 'R*1'. Model RW constitutes an exception as RW1 remains very short (1.8 km, see main text).

Figure 4. It would be good to explain at the beginning of the figure caption that this figure concerns the RT model.

We agree and added a short note.

Your movies are extremely large (some over 1 Gb!); it is certainly possible to create smaller movie-sizes from a set of pictures, and I believe that this is important for readers that do not have a high bandwidth connection.

We agree. All movies are now approx. 5 times smaller with a maximum size of 122 Mb. These smaller videos can be uploaded upon resubmission.
* * *
[Figure]

**Fig. 1.**

horiz. velocity

Time = 110.5 , Timeframe = 545
**Poisson's ratio of 0.125**

slip velocity

plastic strain pes

local friction coeff.

**Fig. 2.**

[Figure]

Fig. 3.

---

## Author Response (AR1)

16th June 2020

Dear Editor,

Thank you for your helpful comments on our manuscript entitled "Characteristics of earthquake ruptures and dynamic off-fault deformation on propagating faults" [Paper se-2020-16]. We are grateful for the constructive and thoughtful comments made by the reviewers. We have addressed their questions, which are quoted below in blue. Text in red indicates text added to the new version of the manuscript. We also provide a PDF version of the revised manuscript in which we highlighted the changes in red (deleted) and blue (added). All line numbers in the letter below refer to the tracked-changes document. We hope that our revised manuscript has clarified the questions raised by the reviewers and made the paper stronger.

Best regards,
Simon Preuss (on behalf of all co-authors)
* * *
**Reviewer 1 - Michele Cooke**

**A.** The assumption of lateral propagation of strike-slip faults does not consider how crustal faults might form by the upward propagation and/or linkage of early fault segments. Experiments of strike-slip fault evolution show upward propagation with the formation of an early set of echelon faults that link to form a through-going strike-slip fault (e.g., Tchalenko, 1970; Hatem et al., JSG 2017). We don't have reason to believe that crustal strike-slip faults would initiate differently from experiment observations. The text uses results of Perrin et al. (2016) that faults are most mature along their centers to justify lateral propagation. Lateral variation in fault maturity don't preclude early upward propagation that would produce echelon segments that may link earlier along some portions of the fault than others. Unlike the quasi-2D simulations in this paper, the base of 3D laboratory experiments distributes shear within the suprajacent material in a manner analogous to crustal systems where mid crustal deformation drives upper crustal faulting. I'm not saying that the investigation of lateral propagation of strike-slip faults within this paper is unreasonable. This is a great first step towards understanding the complex evolution of strike-slip faults but may be only part of the story. To strengthen the implications of the paper, the introduction and discussion of the manuscript should include consideration of the 3D context of these strike-slip faults. How might the findings differ if strike-slip faults initiate with upward propagation followed by linkage?

We thank the reviewer for this thoughtful comment and for pointing out differences between 2D and 3D experiments. Indeed, our 2D (map view) and 2.5D simulations cannot model upward propagation of fault segments. However, there are other scenarios for fault strike-slip formation that are mechanically viable, such as those demonstrated in our paper and

supported by field observations and other lab experiments. Both scenarios are not exclusive: it is very possible that the initiation of a fault is driven from below by the process the reviewer describes, but, as shown here and elsewhere, the subsequent lateral growth does not need to be driven from below.

We have followed a common practice in 2D numerical modeling of fault growth/branching or rupture propagation in strike-slip faults to simulate map view experiments (e.g. Kame et al., 1999; Poliakov et al., 2002; Rice et al., 2005; Meyer et al., 2017; Herrendörfer et al., 2018; Preuss et al., 2019). With our approach we are focusing on a very important part of the problem: the lateral growth stage.

To strengthen and clarify the scope of the paper we add the following to the introduction (line 31) :

Analogue experiments have shown that strike-slip faults can initiate by upward propagation and linkage of an early set of echelon faults to form a through-going fault (e.g. Tchalenko, 1970; Hatem et al., 2017). Further growth towards a through-going strike-slip fault generally occurs due to lateral propagation and the structural fault complexity usually increases towards the younger portions at the fault tip (Perrin et al., 2016a; Cappa et al., 2014).

And we add to the existing text (black italic) in the discussion section (line 688) :

This approximation does not actually account for the third dimension and neglects parameter variations with depth as well as a possible change of the fault dip angle with depth. In this study faults are always vertical in a plane-strain sense, cutting through the entire upper crustal layer. This implies that faults in our models can not initiate at depth and link from an early set of echelon faults that propagate upwards as shown by analogue experiments. *Further, the simulations exclude a temperature-dependent rheology that would imply rheology changes with depth.*

**B.** The conclusions that the strike-slip faults grow predominantly in the aseismic period of the earthquake cycle is based on the assessment that the rate weakening results, with their fast coseismic growth, are unreasonable.

We emphasize that all four reference models have predominantly aseismically growing faults. This was stated at various points in the old version of the manuscript already. The longest seismically grown fault segment in the rate weakening model measures 12.3 km (line 313) and the rest of the newly formed fault (i.e. ~30 km for fault RW2) was produced aseismically. Moreover, the rate weakening model does not have faster and more coseismic growth than the other models. We add that the rate weakening results are not per se considered unreasonable. Our results in section 3.1 show that a bulk rheology with constant rate-sensitivity (including the rate weakening model) favors a faster fault growth of up to 77 km/yr in contrast to more realistic fault growth rates of ~1.8 km/yr in model RT.

To clarify our findings we add to line 483 (beginning of discussion):

Our results suggest that all four reference models have predominantly aseismically growing faults. A bulk rheology with constant rate-sensitivity favors a faster fault growth. In contrast, the heterogeneities introduced by a weakening of the RSF parameters $L$ and $b$ slow down the

faulting process due to the absorption of energy by the weakening mechanism. As a consequence, the faults in the model RT that transition from rate-strengthening to rate-weakening can extend in alternating seismic and aseismic growth periods. Only if the region ahead of the fault tip has experienced distinct plastic strain and $L$ and $b$ are altered to create a rate-weakening fault earthquakes can propagate on there. Otherwise, dynamic rupturing is hindered in the intact bulk, where $L$ is still high and $b$ is still low and rate-strengthening, respectively. This contrast of large $L$ and low $b$ in the bulk rock results in intermittent seismic and aseismic growth sequences. We think this behaviour reflects the natural growth of crustal faults better than constant values of $L$ and $b$, which lead to rapid fault propagation after singular earthquakes.

Furthermore, evolution of $L$ and $b$ with strain was observed in laboratory studies (e.g. Beeler et al., 1996; Scuderi et al., 2017; Marone and Kilgore, 1993).

The differences between the two end-member bulk rheologies have major implications on the dynamics and geometry of fault evolution, which are discussed in this section.

We furthermore add to line 553:
However, in general fault growth predominantly occurs through aseismic deformation in all four reference models, independent of the type of bulk rheological behaviour. That is because a seismic rupture only reaches the current fault tip if this part of the fault is already highly localized. This is solely the case in the first earthquake when the entire fault trace coincides with the predefined, weak, mature fault.

We furthermore add to line 260:
The models with constant rate-sensitivity (models RS, RN, and RW) have fast fault growth rates of up to 77 km/yr that are much faster than in model RT. Despite this major difference, all reference models have in common that fault growth during the earthquakes contributes only a small portion to the total formed fault length FL (Fig. [3b]).

We do have evidence that faults can link during earthquake rupture along previously unmapped segments. This process of quick coseismic linkage could look a lot like the results of the rate weakening models that propagate towards the edges of the model. If the models had a second fault segment, perhaps the rate weakening models would showing linkage of the segments in a very reasonable way. For this reason, perhaps the rate weakening rheology may be overlooked should not be considered completely unreasonable.
We agree that fault linkage can occur during earthquake rupture along previously unmapped segments. We did show this in Figure 9 for model HPT. Adding to that we explained in the previous comment that we do not per se consider the rate weakening rheology as unreasonable.
We are currently running models of a natural earthquake example in which we predefine several unlinked segments that are supposed to link on the long-term (Preuss et al., 2020, EGU General Assembly 2020 presentation: link). However, fault linkage is a complex

process that needs further investigation and is not among the main aspects of this paper, which is already long. We refer to the last two sentences of our manuscript:
*With our work we provide the basis for simulations and analyses of complex evolving fault networks subject to long-term and short-term dynamics. The approach we presented has potential to be applied to a more realistic fault map in a future study.*

**C.** This question is related to the comment of B. Rate neutral and rate weakening are rheologies that we expect in the crust. In this study, they are excluded because the models do not produce more than one earthquake. What model parameters could be altered (for example, in a future study) to get more than one earthquake cycle for the rate neutral and rate weakening models?

We are not sure if rate neutral and rate weakening are the most realistic rheologies we can expect in the crust. Laboratory observations show that RSF parameters *(a-b)* and *L* weaken with plastic strain (Beeler et al., 1996; Scuderi et al., 2017; Marone and Kilgore, 1993). This behavior suggests that a crustal bulk rheology with no rate-sensitivity (like constant rate-neutral or constant rate-weakening) is less likely. We conducted up to ~1000 experiments with different parameter setups but were unable to get to more than one earthquake or distinctly slower fault growth rates in any of those models with no rate-sensitivity.

In contrast a rate sensitivity as observed in laboratory studies is incorporated in model RT, which produces the most realistic fault growth rates and earthquake sequences rather than single earthquakes. Thus, we consider model RT the most realistic model and the rate-transitioning rheology the most likely crustal behaviour.

In a future study we suggest to test the following to get more than one earthquake cycle for the rate neutral and rate weakening models:

- Substantially larger model box with mesh refinement close to the fault
- Strongly misaligned initital fault
- Higher value of background pressure
- Higher initial bulk state variable
- Higher cohesion potentially in combination with strain weakening of cohesion

**D.** Some parameters seem a bit outside of expected crustal ranges. The width of the plastic yielding fan seems large.

We emphasize that observations distinguish between outer and inner damage zones. The "outer damage zone" (Perrin et al., 2016) is much bigger than the "inner damage zone" (Savage and Brodsky, 2011). In our study we furthermore find that the initial orientation of the fault and various parameter(-combinations) alter the width of the plastic yielding fan severely:

In section 3.3 and particularly in Figure 8 we show that the width of the plastic yielding fan depends on the initial orientation of the fault in the surrounding stress field. The large fan width results from the misorientation of the predefined fault. Thus the initial fault orientation

relative to the regional stress is a crucial parameter. A corollary is the possibility to distinguish between optimally oriented and severely misoriented faults by assessing the extent of coseismic off-fault damage.

In section 4.4 we comment on this particularity of our findings. For example we write in line 582:

*Our findings regarding the time-dependent optimality of the fault angle have implications for nature and for future dynamic rupture modeling studies. Active fault strands in nature that are surrounded by severe localized or diffuse damage zones, possibly extending far into the host rock, are strongly misaligned with the interseismic far-field stress field. This misalignment may be increased dynamically during seismic rupturing. This means that individual fault traces may reflect the local geology, structure or stress state rather than the prevailing far-field, long-term stress field and this effect would vary from segment to segment randomizing the fault pattern (Moore and Byerlee, 1989). This explains the complex nature of inter-branched crustal fault systems.*

We add the following to the new version of the manuscript (line 669):

The width of the plastic fan in our models is larger than that seen in observations from Savage and Brodsky (2011). This difference is related to the non-optimally oriented fault of the reference model, which was discussed in section 4.4 and which is compared to an optimally oriented fault with a significantly lower width of the plastic yielding fan in figure 8. Furthermore, the plane strain assumption in our 2.5-D model assumes a constant thickness of the seismogenic fault with depth-constant rate-weakening behavior, which favours a larger width of the plastic yielding fan generated during earthquakes if it is compared to a natural fault which typically has alternating rate-weakening and rate-strengthening patches. Additionally, in our model the width of the fan is controlled by several parameters, of which the thickness of the elastic layer $T$ on top of the visco-elastic half-space has the greatest impact. Higher values of *(a − b)* as well as high and low initial bulk host rock state variable values $\theta hr$ decrease the fan width significantly.

The static frictional strength of the faults of 0.6 is high when we consider that crustal faults have fluids.

We agree with the reviewer on this point. However, we assume a typical pore fluid pressure ratio of $\lambda \sim 0.67$ that increases the background pressure to a lithostatic pressure of *PBlith =60.6MPa* (line 221). Alternatively, the pore fluid pressure could be included to calculate an effective friction coefficient of 0.6 * (1 - 0.67) = 0.2 at shallower depths  This difference in perspectives reflects procedures in different communities. The geodynamics community usually defines an effective friction coefficient by multiplying the fluid pressure ratio with friction, whereas the seismicity community leaves the friction coefficient untouched and works with effective stress or pressure (stress minus fluid pressure). In addition, due to the low initial state on the predefined fault the effective friction coefficient on this fault drops to values around $\mu \sim 0.4$ as the fault is ruptured.

The 20 km choice for maximum fault zone width for the heuristic fault zone thickness, needs stronger substantiation.

We add to line 180:

The upper fault width limit Wmax is defined as the width of inelastic interseismic deformation obtained from fault-parallel GPS and InSAR data. We get a first order estimate of this quantity by measuring the half width of the fault-parallel velocity approaching the far-field plate velocity asymptotically. Wmax can vary significantly between ~ 2 km (Jolivet et al., 2013) and ~ 100 km (Jolivet et al., 2015; Lindsey and Fialko, 2016) in natural faults and depends on the crustal material and thickness, the rate of deformation and the size of the respective fault zone. Consequently, we set Wmax = 20 km as an averaged proxy for the fault width in the interseismic phase. The relation 12 can be interpreted as a heuristic fix to the problem of grid-size-dependent localization in continuum models with RSF.

**E.** Something to consider in the discussion of the paper is the role of nearby faults on the 'bending' of fault traces. While fault strike may bend in response to changing slip conditions, most crustal faults develop within a complex system where they might interact with nearby faults. Slip on a nearby fault (such as the Garlock fault near the San Andreas) may in many cases have larger impact on the bending of faults than difference in aseismic and coseismic lateral propagation.

This is very right and we agree with the reviewer on these points. Indeed, they were addressed in our manuscript already. Especially in section 3.4, in which we introduce model HPT, we noticed and mentioned fault bending and interaction (line 462):

*In the following we analyze several indications of fault and rupture interactions due to stress changes that are typically ignored in seismic cycle models. They include: (1) Rupture arrest when two sub-parallel ruptures get too close to one another. This can be observed for fault HPT001, which stops growing because the stresses on the extensional side of the subsequently forming branch HPT01 increase, get dominant and limit the compressional side stresses of HPT001. As a consequence, only extensional stresses remain at the tip of HPT001, such that the fault gets thinner on its compressional side (Fig. [9c,b]). This leads to (2) a stop in fault growth and fault abandoning.* **Further, fault bending (3) is observed as fault HPT02 approaches HPT0001 and the former starts to bend due to local interactions of stresses. After bending, both faults intersect (4), which causes HPT02 to terminate** *(5). All together, this behavior is well visible in the video of the HPT simulation, 19 Mb. Consequently, new interjacent branches can stop if their extensional side stress field interacts with the compressional side stress field of another rupture. This is the case when the branches of two subparallel ruptures get close to one another. In this process, the fault on the extensional side is likely to continue extending. This line of reasoning applies for a dextral fault system and is reasonable since the evolving fault structure as a total has an extensional character, which*

*means that an extensional stress state is predominant and the extensional fault's side is favored.*

We have summarized our findings and have discussed them in section 4.5 (line 608):
*The single main fault rupture in this model excites 10 dynamic secondary ruptures on the extensional side bulk that can arrest, bend, converge, intersect and get abandoned. This complexity is linked to variations of the normal stress during and between earthquake sequences, which affect the evolving fault pattern. That behavior highlights the importance to include a varying normal stress in earthquake cycle models instead of assuming a constant normal stress. ...*

`An interesting feature in model HPT is the main fault replacement (or jump). This is reflected in the singular growth and slip activity of the outermost fault branch HPT0001 at the end of the simulation. In the dynamically altered stress field this outermost fault branch is most favorably oriented. A main fault jump was reported in southern California where the San Gabriel fault was originally the main strand of the San Andreas fault, but was replaced at about 4 Ma (Moore and Byerlee, 1991). Faults that are unfavorably oriented for large amounts of slip will be replaced by progressively better oriented faults (Moore and Byerlee, 1989). Fault branch interaction occurs also on the long-term when the stress fields of approaching fault strands start to interfere (manifest in a seismically initiated incipient connection between RT1 and RT2 at x~140 km in Fig. 3). Seemingly, the fault system intends to increase its efficiency by decreasing the fault complexity on the long-term due to fault interaction which can lead to abandoning of abundant fault strands. This is another indication, apart from the previously discussed one, that the fault optimizes its growth efficiency and aims at reaching a steady state on the long-term in which seismic and aseismic growth preferentially happen in the same direction.`

Additionally, immature faults may develop bends in their earliest stages when neighboring segments that are not colinear link up to form a single fault surface (Hatem et al., JSG 2017).
We have reported this behavior in the submitted version of the manuscript, already (line 362):
*Additionally, faults RT1 and RT2 interact with each other. Fault RT2 starts to bend towards RT1 at 360 years. This behavior is not recorded in the other reference models. Visible is also a seismically initiated connection between RT1 and RT2 at x=130 km that starts at 355 years.*

**F.** The manuscript strives to address a wide range of conditions/questions. I wonder if some parts of the manuscript, such as the HPT models, might be best served as supplemental material.

We would not want to exclude or shift a part that answered the remarks of reviewer 1 posed under previous point **E**.

**Specific comments**

The paper is very well written. I have a few specific comments that may strengthen the writing in places.

Throughout the manuscript (eg. Line 102, 298 and many others): 'Fault extension' reads a bit odd since extension is a strain term. The text might be clearer with use of 'fault propagation'.
We changed that everywhere.

Line 40 (and there abouts): The use of Riedel terminology for splay fractures strikes me as a bit odd because we typically refer to Riedels as the early formed echelon fault segments. The fractures within the damage zone are more commonly called splay cracks. You may find papers by Cooke (JGR 1997) and Willemse and Pollard (1998) helpful because they show the range of orientation of splay fractures that can develop with different conditions on the fault.
We agree. We changed the respective paragraph and added the reference to Cooke (1997).

Line 51: Define SCEC.
We defined it in the new version of the manuscript.

Line 213: Mixing strain (extensional) and stress (compressional) term. Make these both either strain or stress.
We did not find the term "stress" between lines 211-216. What is the reviewer referring to?

Line 252: spelling of strengthening
We changed that.

Line 253: 'This results' is ambiguous. This what? Being more clear will help the reader. Line 254: Better than what?
We changed that to (line 496):
This contrast of large $L$ and low $b$ in the bulk rock results in intermittent seismic and aseismic growth sequences. We think this behaviour reflects the natural growth of crustal faults better than constant values of $L$ and $b$, which lead to rapid fault propagation after singular earthquakes. This difference has major implications on the dynamics and geometry of fault evolution, as discussed in the next section.

Line 399 and throughout: I'm not a fan of the acronym OOF for optimally oriented fault model. Why not just spell it out since you already have a lot of acronyms and only use OOF for one section of the paper?

The term OOF is now spelled out in the new version of the manuscript.

Line 497: This is just one paper, for which there is a rich literature. Add e.g. and some more citations.

We added (line 525):

Stress analysis of a crack loaded in mode II explains the formation of tensile fractures at the crack tip (e.g. King and Sammis, 1992; Cooke, 1997; Poliakov et al., 2002; Rice et al., 2005).

Line 540: Another interesting study is a quasi-static dynamic model of Savage and Cooke (JSG 2010). That study differs from the ones cited in that it does not limit damage development along pre-existing mesh. So, the results of Savage and Cooke (JSG 2010) might be interesting to compare to your new model results.

We added (line 572):

The value of the slip-weakening distance was shown to regulate between more continuous along-strike damage and concentrated fracturing at fault tips (Savage and Cooke, 2010).

Line 573-574: This is an interesting result. I believe that this finding confirms results of Jiang and Lapusta – it could be helpful to cite their work here.

Jiang and Lapusta (2016, 2017) use a depth dependent effective normal stress. However, our models feature, on top of a spatially varying effective normal stress (in our case effective pressure), a temporal variation of the effective normal stress (effective pressure). To make it more clear we add to line 609:

This complexity is linked to spatial and temporal variations of the normal stress during and between earthquake sequences, which affect the evolving fault pattern. That behavior highlights the importance to include both spatially and temporally varying normal stress in earthquake cycle models instead of assuming a constant normal stress or only assuming a depth dependent normal stress.

Line 583: There is a rich literature on the development of new faults that are more efficient. Add e.g to this reference.

Thanks, we added that.

Line 594: Cooke (JGR 1997) show that changes in friction distribution near fault tips alters the stress concentration and the angle of the splay crack. The change in friction arises in the transition between mature fault with static friction to immature fault with higher friction. Could this process be contributing to your observation of changing splay angle?

A change in the splay angle occurs due to earthquakes (as discussed in section 4.2) and consequently due to differences between aseismic and seismic fault growth (as discussed in section 4.3). Both depend on the optimality of the pre-existing fault angle (as discussed in section 4.4) and are influenced by fault branch interactions (as discussed in section 4.5). As shown in Preuss et al. (2019) the local friction coefficient at the tip of an aseismically and a

seismically growing fault differ substantially, which leads to different fault orientations. So yes, the transition between a mature fault with a given friction to an immature fault with a different (possibly higher) friction is contributing to the observation of changing splay angles. This finding is extensively mentioned throughout the manuscript and in Preuss et al. (2019). Hence, we do believe a further explanation or interpretation is not needed in section 4.6.

Line 602: Add Cooke (JGR 1997) to this reference list as it is very much related to these other good papers.
Thanks, we added that.

Line 628: The width of the plastic fan in the models is larger than that seen in Savage and Brodsky (2011).
We agree and refer to our answer of the reviewers remark **D** in which we discuss that the size of the plastic off-fault fan is a result of a misorientation of the predefined fault in the reference models. As shown in Figure 8, an optimal oriented fault results in a severe decrease of the plastic fan. To make this clearer we add to line 669:
The width of the plastic fan in our models is larger than that seen in observations from Savage and Brodsky (2011). This difference stems from the non-optimally oriented fault of the reference model, which was discussed under 4.4 and which is compared to an optimally oriented fault with significantly lower off fault yielding in figure 8. Additionally, in our model the width of the fan is controlled by several parameters, of which the thickness of the elastic layer $T$ on top of the visco-elastic half-space has the greatest impact. Too high values of $(a - b)$ and too high and too low initial bulk host rock state variable values $\theta hr$ decrease the fan width significantly.

**Reviewer 2 - Boris Kaus**

**1.1) Yield stress criteria**
You correctly write that at yield F=0, and you employ a standard Drucker-Prager yield function (eq. 9). Yet, your expression for the yield function (eq. 8) is incorrect, which can be best illustrated graphically:

[Figure]

The plot shows the yield function (black) together with the Mohr-Circle (green circle, which has radius $_{II}$). At yielding (F=0), the Mohr-Circle exactly touches the yield stress function. This condition is not $_{II} = $ $_{yield}$ (your eq. 4), as this gives the red circle (which predicts a stress that is somewhat larger than the yield stress). Instead, we can use trigonometry to compute the condition for F=0. If we define the effective angle of friction $ $ as:
tan $ = $ $_1(1 - )$,
we can define the yield condition (F=0) as: $_{II} = $ $sin( ) + $ $cos( )$

Since this is correctly described in the textbook of one of the co-authors of this manuscript, I suspect that it is incorporated correctly in the software. I also don't know how big an effect it will make on the results, even if this would not be the case (to be tested). Yet, in any case, it would be good if you can correct your description.

We thank the reviewer for this important comment and a clarifying discussion, which together helped improving the paper.
We clarified the yielding function description and changed notations to avoid confusion. We use a modified Drucker-Prager yielding condition with constant compressive strength and variable friction coefficient.
We changed the method description in line 141:
The onset of plastic deformation is defined by the yield criterion:

$F = \tau II - \sigma c - \mu eff(RSF)\ Peff,$

where $Peff = P - Pfluid = P\ (1 - \lambda)$ with the pore fluid pressure factor $\lambda = P\ /Pfluid$ , $\sigma c$ is the constant compressive strength that marks the residual strength at $P = 0$ and $\mu eff(RSF)$ is a variable effective friction parameter that we define based on our continuum

**RSF formulation. We use a modified Drucker−Prager plastic yield function (Drucker and Prager, 1952) in the form:**

$\sigma_{yield} = C(RSF) + \mu(RSF)\, P_{eff}$,

where

$\mu(RSF) = \tan(\sin^{-1}(\mu_{eff}(RSF)))$ is the local friction coefficient that is widely used and obtained form laboratory experiments

and

$C(RSF) = \sigma_c / \cos(\sin^{-1}(\mu_{eff}(RSF)))$ is the local cohesion.

The local effective friction parameter $\mu_{eff}(RSF)$ evolves according to the invariant reformulation of rate- and state-dependent friction for a continuum, introduced by Herrendörfer et al. (2018). This formalism was applied to freely and spontaneously growing seismic and aseismic faults by Preuss et al. (2019), by interpreting how plastic deformation starts to localize and forms a shear band that approximates a fault zone of finite width that can host earthquakes. Localized bulk deformation and fault slip are related by defining the plastic slip rate $V_p$ as

$V_p = 2\dot{\varepsilon}_{II(p)} W$,

where W denotes the width of the fault zone in the continuous host rock. We formulate $\mu_{eff}(RSF)$ as:

$\mu_{eff}(RSF) = a\, \text{arcsinh}\, (V_p/2V_0\ \exp((\mu_0 + C/P + b\ln(\theta V_0/L))/a))$,

where a and b are laboratory-based, empirical RSF parameters that quantify a direct effect and an evolution effect of friction, respectively, L is the RSF characteristic slip distance, $\mu_0$ is a reference friction coefficient at a reference slip velocity $V_0$ (Lapusta and Barbot, 2012), and C is the cohesion as part of the state variable $\theta$ (Marone et al., 1992) that evolves according to the aging law:

$d\theta/dt = 1 - V_p\theta/L$.

Additionally, we updated Figure 4 according to the recomputed relative fault angles based on equation 9.

**1.2) Elastic material parameters**

Your choice of having the same values for bulk and shear moduli (table 1; both 50 GPa) results in a Poisson ratio of 0.125. That might be appropriate for already damaged rocks, but perhaps not so much for intact rocks. How sensitive are your results to the particular choice of Poisson ratio?

We thank the reviewer for this question. A Poisson's ratio of 0.125 is on the lower end of values for rocks, but still common for a wide range of rocks as for example shown in *Poisson's ratio values for rocks* (H. Gercek, 2007). Furthermore, we tested a range of shear moduli resulting in varying Poisson's ratios and found only a marginal impact on the model results. In particular, the main messages of our manuscript are not influenced by changes in the Poisson's ratio. We illustrate this by comparing the snapshots of simulations with Poisson's ratio of 0.125 and 0.25. We focus on the dynamically generated off-fault yielding at approximately the same deformation stage just before the rupture hits the end of the predefined fault. Both snapshots are attached. The differences comprise:

- First earthquake nucleates 75 years later for $\nu = 0.25$.
- The maximum slip velocity is ~ 0.08 m/s higher for $\nu = 0.25$.
- The off-fault splay localization is more irregular with a higher degree of localization and a slightly higher off-fault reach for $\nu = 0.25$.

[Figure]

[Figure]

In light of our answer above we add to line 703 of the manuscript:

Our choice of parameters results in a Poisson ratio of 0.125. Such a relatively low Poisson's ratio is on the lower end of values for rocks, but still common for a wide range of rock types as for example shown in Gercek (2007). To illustrate the impact of different Poisson's ratios we have tested a range of different shear moduli resulting in varying Poisson's ratios. These tests have shown that the main messages of our manuscript are not influenced by changes in the Poisson's ratio.

**2) Minor remarks:**

- Table 1: I suppose that the host rock cohesion is 6 MPa, and not 6e6 MPa?
  This is correct. We changed that.

- Fig. 3: you show an overview of several different models. Yet, are the snapshots chosen to have approximately the same plastic strain, deformation stage or time? Would be good to mention it.
  Thanks for this comment. We added the following to the figure caption:
  The snapshots in [a] are chosen to have approximately the same deformation stage with regard to fault length 'R*1'. Model RW constitutes an exception as RW1 remains
  very short (1.8 km, see main text).

- Figure 4. It would be good to explain at the beginning of the figure caption that this figure concerns the RT model.
  We agree and added a short note.

- Your movies are extremely large (some over 1 Gb!); it is certainly possible to create smaller movie-sizes from a set of pictures, and I believe that this is important for readers that do not have a high bandwidth connection.
  We agree. All movies are now approx. 5 times smaller with a maximum size of 122 Mb. These smaller videos can be uploaded upon resubmission.

**Characteristics of earthquake ruptures and dynamic off-fault deformation on propagating faults**

Simon Preuss[1], Jean Paul Ampuero[2], Taras Gerya[1], and Ylona van Dinther[3]

[1]Geophysical Fluid Dynamics, Institute of Geophysics, Department of Earth sciences, ETH Zürich, 8092 Zürich, Switzerland
[2]Géoazur Laboratory, Institut de Recherche pour le Développement - Université Côte d'Azur, Campus Azur du CNRS, 06560 Valbonne, France
[3]Tectonics, Department of Earth Sciences, Utrecht University, Princetonlaan 4, 3584 CB, Utrecht, the Netherlands
**Correspondence:** Simon Preuss (sipreuss@ethz.ch)

**Abstract.** Natural fault networks are geometrically complex systems that evolve through time. The evolution of faults and their off-fault damage pattern are influenced by both dynamic earthquake ruptures and aseismic deformation in the interseismic period. To better understand each of their contributions to faulting we simulate both earthquake rupture dynamics and long-term deformation in a visco-elasto-plastic crust subjected to rate-and-state-dependent friction. The continuum mechanics-based numerical model presented here includes three new features. First, a 2.5-D approximation to incorporate effects of a viscoelastic lower crustal substrate below a finite depth. Second, we introduce a dynamically adaptive (slip-velocity-dependent) measure of fault width to ensure grid size convergence of fault angles for evolving faults. Third, fault localization is facilitated by plastic strain weakening of bulk rate-and-state friction parameters as inspired by laboratory experiments. This allows us to for the first time simulate sequences of episodic fault growth due to earthquakes and aseismic creep. Localized fault growth is simulated for four bulk rheologies ranging from persistent velocity-weakening to velocity-strengthening. Interestingly, in each of these bulk rheologies, faults predominantly localize and grow due to aseismic deformation. Yet, cyclic fault growth at more realistic growth rates is obtained for a bulk rheology that transitions from velocity-strengthening friction to velocity-weakening friction. Fault growth occurs under Riedel and conjugate angles and transitions towards wing cracks. Off-fault deformation, both distributed and localized, is typically formed during dynamic earthquake ruptures. Simulated off-fault deformation structures range from fan-shaped distributed deformation to localized  splay faults. We observe that the fault-normal width of the outer damage zone saturates with increasing fault length due to the finite depth of the seismogenic zone. We also observe that dynamically and statically evolving stress fields from neighboring fault strands affect primary and secondary fault growth and thus that normal stress variations affect earthquake sequences. Finally, we find that the amount of off-fault deformation distinctly depends on the degree of optimality of a fault with respect to the prevailing but dynamically changing stress field. Typically, we simulate off-fault deformation on faults parallel to the loading direction. This produces a 6.5-fold higher off-fault energy dissipation than on an optimally oriented fault, which in turn has a 1.5-fold larger stress drop. The misalignment of the fault with respect to the static stress field thus facilitates off-fault deformation. These results imply that fault geometries bend, individual fault strands interact and that optimal orientations and off-fault deformation vary through space and time. With our work we establish the basis for simulations and analyses of complex evolving fault networks subject to both long-term and short-term dynamics.

**1 Introduction**

Immature strike-slip faults accumulate displacement over time as they undergo a slip localization process. On the long-term, these structures can become deeply penetrating, major faults that represent a highly localized weak zone through the lithosphere (Norris and Toy, 2014). The majority of slip is thereby confined to the cores of the principal faults (Chester et al., 1993). Most prominent examples, like the San Andreas and the North Anatolian fault systems, span lengths of several hundreds of km (e.g., Sibson, 1983).  Analogue experiments have shown that strike-slip faults can initiate by upward propagation and linkage of an early set of echelon faults to form a through-going fault (e.g. Tchalenko, 1970; Hatem et al., 2017). Further growth towards a through-going strike-slip fault generally occurs due to lateral propagation and the structural fault complexity usually increases towards the younger portions at the fault tip (Perrin et al., 2016a; Cappa et al., 2014). In this area diverse fault patterns and fault networks are found. Analog experiments, structural geology and fracture mechanics define a variety of different secondary fault structure types: branching faults, one-sided horsetail splay faults, synthetic and antithetic Riedel  faults, splay cracks, wing cracks and mixed modes  (e.g. Cooke, 1997; Hubert-Ferrari et al., 2003; Kim et al., 2004; Mitchell and Faulkner, 2009; Aydin and Berryman, 2010; Perrin et al., 20 . The different types mainly differ in fault angle and fracture mode. For example, R Riedel  shears that form in response to Coulomb failure make an angle of $10° - 20°$ with the main fault (Riedel, 1929; Logan. et al., 1979; Logan et al., 1992), while opening-mode wing cracks grow in the direction of the most tensile circumferential stress and hence have a greater angle (Erdogan and Sih, 1963; Ashby and Sammis, 1990). The terms 'shear fracture', 'splay' and 'splay fault' are used as equivalents in this study. R1 refers to synthetic Riedel shears and R2 refers to antithetic conjugate Riedel shears often also named R'.

[Figure]

[a]   Off-fault fan

[b]   Synthetic R1-Riedel horsetail splays

[c]   Off -fault fan, R1-splays and R2-splays

[d]   Wing crack or T-tension fracture

[revised manuscript text omitted]

Le Pichon, X., Chamot-Rooke, N., Rangin, C., and Sengör, A. M. C.: The North Anatolian fault in the Sea of Marmara, Journal of Geophys-
ical Research: Solid Earth, 108, 1–20, https://doi.org/10.1029/2002jb001862, 2003.

Lee, S., Reber, J. E., Hayman, N. W., and Wheeler, M. F.: Investigation of wing crack formation with a combined phase-field and experimental
approach, Geophysical Research Letters, 43, 7946–7952, https://doi.org/10.1002/2016GL069979, 2016.

Lehner, F. K., Li, V. C., and Rice, J. R.: Stress Diffusion Along Rupturing Plate Boundaries, Journal of Geophysical Research, 86, 6155–6169,
1981.

Lindsey, E. O. and Fialko, Y.: Geodetic constraints on frictional properties and earthquake hazard in the Imperial Valley, Southern California,
Journal of Geophysical Research B: Solid Earth, 121, 1097–1113, https://doi.org/10.1002/2015JB012516, 2016.

[revised manuscript text omitted]